# LIMA (v2.0): A full two-moment cloud microphysical scheme for the mesoscale non-hydrostatic model Meso-NH v5-6

Marie Taufour[1], Jean-Pierre Pinty[1], Christelle Barthe[1], Benoît Vié[2], and Chien Wang[1]

[1]Laboratoire d'Aérologie, Université de Toulouse, UT3, CNRS, IRD, Toulouse, France
[2]Centre National de Recherches Météorologiques - UMR 3589, Toulouse, France

**Correspondence:** Christelle Barthe (christelle.barthe@cnrs.fr)

**Abstract.** A full two-moment microphysics parameterization of the LIMA scheme (LIMA standing for Liquid Ice Multiple Aerosols, and hereafter named LIMA v2.0) has been developed and successfully implemented in the Meso-NH cloud-resolving model. The novelty of the scheme is a set of prognostic equations of the number concentration of each precipitating ice category (snow-aggregates, graupel and hail), in complement to the prediction of the mass mixing ratios. As a result, new microphysical conversion rates are introduced and explicitly computed using the size distributions of the hydrometeors.

The new LIMA v2.0 scheme has been tested for an idealized deep convection case against the original LIMA scheme characterized by an empirical number concentration-mixing ratio relationship applied to the precipitating ice. Inclusion of number concentration equations for the snow-aggregates and graupel significantly alters the microphysical structure and dynamical evolution of the simulated supercell. When comparing to the results obtained with the previous version of LIMA, the new v2.0 of the scheme tends to increase the pristine ice mixing ratio, to decrease the other ice hydrometeors by slowing down the growing processes of snow/aggregates, graupel and hail, and to enhance the feedbacks between raindrops and the ice phase hydrometeors. This comparison also emphasizes the unreasonable diagnostic approach used to estimate the number concentration of precipitating ice particles in the previous version of the scheme. The new scheme is more efficient at producing earlier raindrops at ground level and reducing hail precipitation.

## 1 Introduction

Cloud microphysical processes play a central role in determining the life cycle of the clouds, the formation of precipitation as well as the radiative transfer in cloudy environment. The response of these processes to the variation of aerosol fields under different dynamical and thermodynamical backgrounds is also essential in estimating the impacts of natural and anthropogenic aerosols on Earth's radiative budget, water cycle, and atmospheric circulation as well as weather patterns (e.g., Lau et al., 2006; Christensen and Stephens, 2011; van den Heever et al., 2011; Tao et al., 2012; Goren and Rosenfeld, 2012; Lee, 2012). An adequate representation of cloud microphysical processes in climate and weather models is thus essential for high quality projections.

Laboratory studies (e.g., Simpson, 1983; Schnaiter et al., 2016; Simpson et al., 2018) alongside real-world case studies (e.g., Heymsfield et al., 2002, 2010, 2013; Delanoë et al., 2005, 2014) have long sustained the advancements of knowledge

that benefit to the development of many microphysical schemes. However, these developments still lack of consensus due to the fact that some microphysical features can be described with more or less sophistication in the limit of the computational resources. In addition, uncertainties also remain in the current numerical schemes particularly those concerning ice phase processes (McFarquhar et al., 2017).

Current microphysics schemes can be categorized into two major types: i.e., the bin schemes (e.g., Hall, 1980; Khain et al., 2000) characterized by discretized size distributions with tens of state prognostic variables, and the modal schemes (e.g., Walko et al., 1995; Pinty and Jabouille, 1998; Thompson et al., 2004), where only one, two or even three moments of the particle size distributions (PSD) are deemed to be sufficient to describe the microphysics states. Owing to their computational efficiency, the moment schemes with different degrees of complexity have been mostly used in a vast majority of applications with a focus on cloud dynamics or microphysics-dynamics interactions. For example, Straka and Mansell (2005) developed a single-moment bulk scheme using mass as the sole prognostic variable, subdividing icy hydrometeors into ten categories according to their shape, size and density, including columnar or wafer-shaped ice crystals, frozen cloud water droplets, snow, three categories for graupel of different densities, frozen drops, and two types for graupel separated by their size. This type of scheme with multiple precipitating ice categories is an attempt to simulate a wide range of storms with limited parameter tuning. Nevertheless, Srivastava (1978) highlighted the advantages of adopting two prognostic moments in determining the PSD of hydrometeors in modeling microphysical processes. Two-moment microphysical schemes have then been developed, allowing more degrees of freedom and thus a better representation of the complex microphysical processes. In such schemes, the number concentration alongside the mixing ratio are commonly selected as the two prognostic moments. Early works chose to represent only one species with 2-moment method, for example the cloud water (Nickerson et al., 1986) or the primary ice crystals (Cotton et al., 1986), while others developed a 2-moment representation for only a part of the hydrometeors, leaving the rest for single moment representation (e.g., Ziegler, 1985; Ferrier, 1994; Vié et al., 2016). This strategy perhaps also reflects the concern about developing excessively sophisticated cloud microphysics schemes and the need to include a variety of aerosol properties in modeling cloud microphysics and evolution. The first full 2 moment schemes (Murakami, 1990; Wang and Chang, 1993; Meyers et al., 1997) were developed to allow a more realistic representation of the cloud systems when compared to radar observations. Subsequently, microphysical parameterizations with 2 or even more moments have become more popular (Reisner et al., 1998; Cohard and Pinty, 2000a; Khairoutdinov and Kogan, 2000; Seifert and Beheng, 2006b; Morrison et al., 2009, and many others) and 2 moment schemes are part of the state of the art in cloud modeling at high resolution. To be complete, it is worth mentioning the P3 (Predicted Particle Properties) scheme (Morrison and Milbrandt, 2015; Milbrandt et al., 2021) where emphasis is put on the prediction of particle properties compared to the more classical approach based on predefined water and ice categories and providing conversion rates. Here our choice is to keep the classical point of view, based on predefined categories, in order to test an improvement of the whole scheme by considering a set of prognostic equations describing the number concentrations of precipitating ice particles.

In recent years, numerous studies have demonstrated the contribution of multi-moment parameterizations in making significant improvement of models' performance. Good examples include better parameterized processes such as sedimentation (taking into account size sorting, Milbrandt and Yau, 2006; Dawson et al., 2010), or evaporation processes with feedback to

updrafts or cold zones under systems (Seifert and Beheng, 2006a; Morrison et al., 2009; Noppel et al., 2010). Multi-moment schemes provide greater variability and precision in hydrometeor size and reflectivity, enabling better comparison with available observations. Seifert et al. (2006) have shown that for an idealized supercell case, the results with a 2-moment scheme are similar to those obtained with a bin-type parameterization. Nevertheless, these multi-moment schemes also require the developers to check more carefully the shape of non spherical ice hydrometeors (related to the fall speed or the density of the hydrometeors). The choice of the parameterization to describe the microphysical processes can also impact the results. Indeed, the conclusions on the impact of aerosol populations on simulated total precipitation vary according to the type of system (Seifert and Beheng, 2006a), but are also strongly influenced by the different microphysical schemes, more than by the addition of moments to describe the hydrometeors in the same scheme (Milbrandt et al., 2010).

The development of a fully 2 moment microphysics scheme still raises the concern of the true benefit of including a set of prognostic equations for each icy hydrometeor concentration. One difficulty is to set the proper boundaries of the graupel PSD for integration over the particle sizes. More importantly, a true difficulty is to provide a robust and consistent parameterization of the fundamental conversion processes that transfer both mixing ratio and number concentration from one category of ice hydrometeor to another one. The snow-aggregate particle case is probably the easiest case to treat because it is similar to that of the raindrops which are generated by the autoconversion (coalescence) of the cloud droplets. According to Harrington et al. (1995), an estimate of 150 μm is an acceptable size limit between the pristine ice crystals and the snow-aggregate category. This value corresponds more or less to the size that small ice crystals should grow by water vapor deposition before experiencing riming by the cloud droplets or ice crystal self-aggregation (Pruppacher and Klett, 1997). The onset of pristine ice riming depends on the crystal habit but the mean size of 150 μm seems to well cover most of the cases (Wang and Ji, 2000). So the growth-conversion scheme of Harrington et al. (1995) can be fully applied to calculate the rates of both mass mixing ratio and number concentration to feed the snow-aggregate particles. In contrast, there are two generating sources of graupel particles in the LIMA scheme: the raindrop freezing after contact with a small ice crystal and the intense riming of the snow-aggregate hydrometeors. In the first case, a tendency of the graupel number concentration can be computed because number concentrations of the raindrops and pristine ice crystals are locally available. Conversely, the expression of a number concentration tendency for the riming-conversion process of the aggregates depends on the way "light" and "heavy" riming intensities are formally defined. All these questions are examined in the next section.

This work describes a full 2 moment microphysical scheme for mixed phase clouds developed for the non-hydrostatic mesoscale model Meso-NH (Lafore et al., 1998; Lac et al., 2018). The original microphysics scheme, the Liquid, Ice, Multiple Aerosols (LIMA) module (Vié et al., 2016) is partially based on a 2-moment representation of the liquid-phase hydrometeors following the original warm microphysics scheme of Cohard and Pinty (2000a). In this study, a full 2 moment version of the original LIMA v1.0 scheme (Vié et al., 2016) has been developed. It is thereby able to describe cloud, rain and all ice hydrometeors using two moments of the size distribution, allowing a better consistency in the treatment of the physical processes, in particular those related to ice phase processes (Vié et al., 2016; Hoarau et al., 2018a). The new version of the LIMA scheme, hereinafter referred to as LIMA v2.0, is described in the first section of the paper, with a focus on the treatment of the snow-aggregate and graupel number concentration tendencies. The second section is dedicated to different behaviors of the two

versions of LIMA in simulating the same idealized supercell case. Finally a conclusion is drawn on the specific properties of the full 2 moment microphysics scheme LIMA, in particular the advantage to reinforce the decoupling between mixing ratios and number concentrations to offer more flexibility to the simulation of cloud evolution.

## 2 Model description

### 2.1 Short overview of Meso-NH

The regional scale model Meso-NH (Mesoscale Non-Hydrostatic, Lafore et al., 1998; Lac et al., 2018) was initially jointly developed by Laboratoire d'Aérologie and Centre National de Recherches Météorologiques in Toulouse (France). A detailed description of the successive versions of the code is available in the scientific documentation of the model (http://mesonh.aero.obs-mip.fr). Meso-NH is a high resolution limited area research model for the simulation of idealized cases (selected cases to study certain aspects of the modeled subject) or the simulation of real meteorological situations over complex terrain with initializa-
tion and coupling data analyses derived from model outputs. Many subgrid scale physical parameterizations are available in the package, such as an EXternalized SURFace submodel (SURFEX, Masson et al., 2013) including a dynamical ocean-wave coupling (Voldoire et al., 2017; Pianezze et al., 2018), deep and shallow convective mass-flux and turbulent schemes (Cuxart et al., 2000; Bechtold et al., 2001; Pergaud et al., 2009), and several microphysics schemes (Cohard and Pinty, 2000a; Vié et al., 2016) coupled to a standard radiative transfer scheme (Morcrette, 1991; Mlawer et al., 1997) and more recently to the
radiation scheme of the ECMWF model (ecRad, Hogan and Bozzo, 2018).

### 2.2 Summary of the LIMA v1.0 microphysical scheme

The scheme described in this paper is an extension of the previous LIMA v1.0 microphysics scheme (Vié et al., 2016). LIMA v1.0 is partially 2-moment with a primary focus on the representation of aerosol-cloud interactions. In detail, LIMA has a 2 moment representation only for the liquid-phase hydrometeors (cloud droplets and rain drops) and for the primary ice crystals.
The whole population of aerosols is categorized into CCN (Cloud Condensation Nuclei) and IFN (Ice Freezing Nuclei) to form droplets and pristine ice crystals, via activation or/and nucleation respectively. LIMA is sensitive to the competition among lognormal PSD parameters and composition of the CCN, and to the lognormal PSD characteristics and solubility of the IFN, as well.

LIMA v1.0 predicts the mass mixing ratio $r$ (the mass of water scaled by the reference mass of dry air $\rho_a$) of six water
species: water vapour, cloud water, rain water, primary ice crystals, snow/aggregates and graupel, and number concentration ($N$) for cloud water, rain water, and ice crystals. Hail is an optional category of hydrometeors depending on the dry/wet growth modes of the graupel.

The PSD for each hydrometeor follows a generalized Gamma distribution:

$$n(D) = Ng(D) = N\frac{\alpha}{\Gamma(\nu)}(\lambda D)^{\alpha\nu}D^{-1}e^{-(\lambda D)^{\alpha}}, \tag{1}$$

where $D$ is the diameter of the particle, $\alpha$ and $\nu$ are free shape parameters (see Table 1 for their values), $g(D)$ the normalized generalized Gamma distribution and $\Gamma()$ the Gamma function (see appendix for a list of symbols). The slope parameter $\lambda$ is a function of the number concentration and the mixing ratio,

$$\lambda = \left( a\frac{N}{r}\frac{\Gamma(\nu + b/a)}{\Gamma(\nu)} \right)^{\frac{1}{b}}. \tag{2}$$

using a mass-size relationship $m(D) = aD^b$ for the liquid and ice condensate, with $a$ and $b$ fixed parameters described in Table 1. The terminal fall velocity $v(D)$ is also related to the particle size $D$ (equivalent to the diameter for the drops):

$$v(D) = cD^d \left( \frac{\rho_{00}}{\rho_a} \right)^{0.4} \tag{3}$$

It takes into account the Foote and Du Toit (1969) correction of the air density with $\rho_a$ the air density, and $\rho_{00}$ the air density at the reference pressure level $P_{00}$. All fixed parameters are described in Table 1.

It can be noticed that the use of a Gamma distribution does not require much computational effort and allows the maximal flexibility. The $p^{th}$ moment of the law ($M(p)$) is written as:

$$M(p) = \int_0^\infty D^p g(D)\, dD = \frac{1}{\lambda^p}\frac{\Gamma(\nu + p/\alpha)}{\Gamma(\nu)}, \tag{4}$$

Then, the mixing ratio ($r$) and the number concentration ($N$) of any hydrometeor are defined respectively by:

$$r = \int_0^\infty m(D)n(D)dD = aNM(b), \tag{5}$$

and:

$$N = \int_0^\infty n(D)dD = NM(0). \tag{6}$$

In LIMA v1.0, the number concentration of snow/aggregates, graupel and hail is estimated using the relationship $N = C\lambda^x$ where $C$ and $x$ are fixed parameters in Table 1. This assumption for aggregates implicitly takes into account the broadening of the particle spectrum to represent coalescence, and also implicitly treats the aggregation process. These conditions are met if: (i) $0 < x$, i.e. roughly reproduce the broadening of spectra (when $\lambda$ decreases) by self-aggregation ($N$ decreases), and (ii) $x < b$, i.e. if the spectrum broadens as the snow/aggregates mixing ratio increases. Compiling results from the literature for snow, graupel, and hail, Caniaux (1983) showed that $C$ and $x$ are linked by an empirical relationship: $log(C) = -3.55x + 7.4$. Then, through sensitivity studies and physical considerations, he determined that the best couple ($C$,$x$) for snow/aggregates was (5,2). However, in the 1-moment ICE3 (Pinty and Jabouille, 1998) and partial 2-moment LIMA v1.0 (Vié et al., 2016) schemes of Meso-NH, $x$ was set to 1 because taking $x$ too close to 2 would lead to some inconsistencies in computing $\lambda$.

The different processes involved in the evolution of the mixing ratio and number concentration of all hydrometeor categories are described in Table 2. The microphysical scheme is sketched in Fig. 1 where each box represents a different category of

water substance. In summary, cloud droplets are initiated following the activation scheme (HENU) as described in Cohard et al. (1998). Once initiated, cloud droplets grow by condensation of water vapour (CND) or evaporate (EVAP) instantaneously to avoid supersaturation over water drops. Then accretion (ACC), autoconversion (AUTO), self-collection (SC) and drops break-up (BU) processes are put in place to initiate precipitating hydrometeors and to make them grow. Raindrops evaporate (EVAP) as they fall below the cloud base. The full 2 moment warm scheme is described in Cohard and Pinty (2000a, b).

Two heterogeneous processes can initiate ice crystals: the formation of ice embryos on insoluble IFN in a supersaturated environment over ice (HIND), and the freezing by immersion of supercooled droplets issuing from partially soluble CCN (IFR). The homogeneous nucleation (HON) takes place when the temperature drops below $-35^\circ$C; it depletes very rapidly the cloud droplets and raindrops. The original IFN nucleation scheme comes from Phillips et al. (2008, 2013) and the adaption to the LIMA constraints (IFN PSD) is given in Vié et al. (2016). In addition, LIMA has three parameterizations of secondary production of ice crystals: the rime splintering mechanism (Beheng, 1987), also known as the Hallett-Mossop process (HM), and the collisional break-up of big ice crystals (CIBU) (Hoarau et al., 2018b) as well as the raindrop shattering when freezing (RDSF) following (Lawson et al., 2015). For the moment, the CIBU and RDSF processes can be activated or deactivated at the user's discretion.

Ice crystals can experience growth by water vapour deposition or sublimation (DEP/SUB) depending on the level of saturation with respect to ice. Pristine ice crystals autoconversion (CNV) forms snow/aggregates. Then, raindrop contact freezing (CFRZ) or heavy riming (HRIM) on the snowflakes is the primary source of graupel. Then a number of interactions between the different hydrometeors are taken into account and listed in Table 2 and Fig. 1. When the temperature is warmer than the triple point temperature ($T_t$), small ice crystals are instantaneously converted into cloud water (IMLT), and snow/aggregates are converted into graupel (CMEL) at a rate proportional to their partial melting following (Walko et al., 1995). Graupels melt by shedding all the liquid water into raindrops (SHED).

When hail category is activated, these particles originate from the graupel category, where the particles likely experience a wet growth mode.

## 3 The new full two-moment version of LIMA

Generally, LIMA v2.0 is an advanced version of the LIMA scheme, the first fully 2 moment microphysics scheme in LIMA family. Version 2.0 inherits the six water species of the LIMA v1.0 alongside their interactions, while it includes additional processes and the prognostic equations of all hydrometeor number concentration in the ice phase. All the processes concerned by the new features described in this section are shown in Fig. 1 and marked with red color.

Basically, LIMA v2.0 is based on v1.0. Specifically, for processes related to snow, graupel and hail already handled in LIMA v1.0, a new prognostic equation is added to the existing routines for handling number concentration transfer rates. For processes newly handled in version 2.0, typically the self-collection of snow, a new routine is created including the parameterization of this process and called up in the LIMA monitor routine. The choice of LIMA version (v1.0 or v2.0, partial or full 2-moment) is made directly in the model namelist. The number of prognostic moments for each hydrometeor type individually is done

thanks to namelist variables which can be set to 1 or 2 (to predict the mixing ratio only, or both the mixing ratio and the number concentration). This newly developed code is included in the official version of Meso-NH starting from version 5-6-0.

## 3.1 Collection/coalescence processes parameterization

### 3.1.1 General formulation

Developing a numerical scheme to adequately simulate the growth of ice particles by collection is a true difficult task. As in many bulk parameterizations, the continuous growth and a simple geometric sweep-out concept for the collection kernel is assumed. The main difference in the treatment of this process in the various microphysical schemes is reflected in how the positive fall velocity differences are handled (see Section 3.1.3).

The collection processes can be categorized in three groups depending on the number of species involved.

(I) In the most general case, a new species $Z$ can be formed during the collection processes ($COL$) involving species $X$ and $Y$ (the less abundant species). This is associated with simultaneous collection and conversion processes, and can be related to conditions on the mixing ratios $r_x$ and $r_y$. So the mixing ratio ($\Delta_{\text{COL}}r_y$) and number concentration ($\Delta_{\text{COL}}N_y$) tendencies of species $Y$ (a loss for $Y$) due to the mass collection of $X$ are:

$$\Delta_{\text{COL}}r_y = \rho_a^{-1} \int_0^\infty \left\{ \int_0^\infty K(D_x, D_y)\, m_y(D_y) n_y(D_y) dD_y \right\} n_x(D_x) dD_x, \tag{7}$$

and:

$$\Delta_{\text{COL}}N_y = \rho_a^{-1} \int_0^\infty \left\{ \int_0^\infty K(D_x, D_y)\, n_y(D_y) dD_y \right\} n_x(D_x) dD_x. \tag{8}$$

The collection kernel $K$ is defined by:

$$K(D_x, D_y) = \frac{\pi}{4}(D_x + D_y)^2 |v_x(D_x) - v_y(D_y)| E_{xy}, \tag{9}$$

with $E_{xy}$ the collection efficiency. The mixing ratio and number concentration tendencies for $X$ (a loss for $X$) are
205 estimated in a similar way, and we can find a similar expression for $\Delta_{\text{COL}}r_x$ and $\Delta_{\text{COL}}N_x$. The mixing ratio tendency of species $Z$ (a gain for $Z$) is the sum of the $X$ and $Y$ losses ($\Delta_{\text{COL}}r_x + \Delta_{\text{COL}}r_y$). The number concentration tendency of species $Z$ is $\Delta_{\text{COL}}N_y$.

(II) When $Z$ is identical to one of the initial species $X$ or $Y$, the collection becomes a two component process, and so only one mixing ratio collection rate needs to be calculated: $\Delta_{\text{COL}}r_y = -\Delta_{\text{COL}}r_x$. In this case, the number concentration of
210 species $X$ or $Y$ varies following Eq.(8).

(III) Collection processes can also be considered as two- or three-component processes when threshold diameters are introduced. For example, when species $X$ is collected by species $Y$, species $Y$ is converted into species $Z$ if and only if the

diameter $D_y$ of $Y$ is greater than a required value $D_y^{lim}$. In this case, only a fraction of species $Y$ (generally the fraction with a diameter greater than the threshold diameter) is converted to species $Z$ and must be removed from category $Y$. The mass of the remaining fraction of species $Y$ increases according to a collection process between the two species $X$ and $Y$ (for instance the riming of aggregates process). So, the growth of $Y$ from $X$ is now:

$$\Delta_{\text{COL}} r_{x \to y} = \rho_a^{-1} \int_0^{D_y^{lim}} \left\{ \int_0^{\infty} K(D_x, D_y) \, m_x(D_x) n_x(D_x) dD_x \right\} n_y(D_y) dD_y, \tag{10}$$

and the growth of $Z$ from both $X$ and $Y$ is:

$$\begin{aligned} \Delta_{\text{COL}} r_{y \to z} &= \rho_a^{-1} \int_0^{\infty} \left\{ \int_{D_y^{lim}}^{\infty} K(D_x, D_y) \, m_y(D_y) n_y(D_y) dD_y \right\} n_x(D_x) dD_x, \\ &= \Delta_{\text{COL}} r_y - \Delta_{\text{COL}} r_{x \to y} \end{aligned} \tag{11}$$

while $\Delta_{\text{COL}} r_y$, the total loss of $Y$ is given by Eq.(7).

The number concentration tendencies are a loss of species $X$ given by Eq.(8), and a loss of species $Y$ equivalent to a gain of species $Z$ following:

$$\begin{aligned} \Delta_{\text{COL}} N_{y \to z} &= -\Delta_{\text{COL}} N_z \\ &= \Delta_{\text{COL}} N_y - \Delta_{\text{COL}} N_{y \to x} \\ &= \rho_a^{-1} \int_0^{\infty} \left\{ \int_{D_y^{lim}}^{\infty} K(D_x, D_y) \, n_y(D_y) dD_y \right\} n_x(D_x) dD_x, \end{aligned} \tag{12}$$

This much more physically-based approach, however, requires a technically more complicated partial integration on the dimensional spectrum of at least one species to calculate mixing ratio trends.

For warm processes (ACC, SC) the Long (1974) parameterization is used (see Cohard and Pinty, 2000a).

### 3.1.2 Cases with particles of different fall speeds magnitude

The collection equations can be simplified when the terminal fall velocity of species $X$ can be neglected in view of that of $Y$ (for instance $X$ represent the pristine ice crystals and $Y$ graupel). Thus, the fall speed of species $X$ in Eq.(9) can be legitimately ignored, and Eq.(7) is developed as follows:

$$\begin{aligned} \Delta_{\text{COL}} r_y &= \rho_a^{-1} \left( \frac{\rho_{00}}{\rho_a} \right)^{0.4} \frac{\pi}{4} a_y E_{xy} \int_0^{\infty} \left\{ \int_0^{\infty} D_y^2 c_y D_y^{d_y} D_y^{b_y} n_y(D_y) dD_y \right\} n_x(D_x) dD_x, \\ &= \rho_a^{-1} \left( \frac{\rho_{00}}{\rho_a} \right)^{0.4} \frac{\pi}{4} E_{xy} c_y \times \int_0^{\infty} a_y D_y^{2+d_y+b_y} n_y(D_y) dD_y \times \int_0^{\infty} n_x(D_x) dD_x, \\ &= \rho_a^{-1} \left( \frac{\rho_{00}}{\rho_a} \right)^{0.4} \frac{\pi}{4} E_{xy} c_y \times a_y N_y M(2 + d_y + b_y) \times N_x, \end{aligned} \tag{13}$$

and Eq.(7) adapted for species $X$ becomes:

$$
\begin{aligned}
\Delta_{\text{COL}} r_x &= \rho_a^{-1} \left(\frac{\rho_{00}}{\rho_a}\right)^{0.4} \frac{\pi}{4} E_{xy} c_y \times \int_0^\infty D_y^{2+d_y} n_y(D_y) dD_y \times \int_0^\infty a_x D_x^{b_x} n_x(D_x) dD_x, \\
&= \left(\frac{\rho_{00}}{\rho_a}\right)^{0.4} \frac{\pi}{4} E_{xy} c_y \times N_y M(2+d_y) \times r_x.
\end{aligned}
\tag{14}
$$

Similarly, for concentrations, we get:

$$
\begin{aligned}
\Delta_{\text{COL}} N_x &= \Delta_{\text{COL}} N_y, \\
&= \rho_a^{-1} \left(\frac{\rho_{00}}{\rho_a}\right)^{0.4} \frac{\pi}{4} E_{xy} c_y N_y M(2+d_y) N_x, \\
&= \frac{N_x}{r_x} \Delta_{\text{COL}} r_x
\end{aligned}
\tag{15}
$$

In many calculations, the fall velocity of ice crystals or cloud droplets is relatively small and thus can be neglected. This is the case for:

1. **Raindrop contact freezing (CFR)** where the collection efficiency is fixed to $E_{ir} = 1$.

2. **Ice crystals aggregation (AGG)** where the collection efficiency is $E_{is} = 0.25 \ e^{0.05(T-T_t)}$ based on Kajikawa and Heymsfield (1989). This is consistent with the decrease of the sticking efficiency of the interacting solid crystals when the temperature is cooler than the water triple point temperature $T_t$.

3. **Partial riming of the cloud droplets (RIM)** where the approach of Farley et al. (1989) is used with the assumption that a conversion of aggregates into graupels may occur for riming aggregates of size larger than $D_s^{lim} = 7$ mm. Thus, the change rates of the cloud droplet, ice crystal and graupel mass and number concentration by riming (RIM) are estimated using Eq.(10-12) with an efficiency $E_{cs} = 1$.

4. **Graupel dry growth (DRYG)** is the sum of individual collection processes that is:

$$
\Delta_{\text{DRYG}} r_g = \Delta_{\text{DRYG}} r_c + \Delta_{\text{DRYG}} r_r + \Delta_{\text{DRYG}} r_i + \Delta_{\text{DRYG}} r_s.
\tag{16}
$$

where $\Delta_{\text{DRYG}} r_c$, $\Delta_{\text{DRYG}} r_r$, $\Delta_{\text{DRYG}} r_i$, $\Delta_{\text{DRYG}} r_s$ and $\Delta_{\text{DRYG}} r_g$ are the mixing ratio tendency of cloud droplets, raindrops, ice crystals, snow/aggregates and graupel, respectively, associated with graupel dry growth. While the graupel number concentration is held constant during this process, the number concentration of cloud droplets, rain drops, ice crystals and snow/aggregates decreases. Since the terminal fall speed of cloud water and ice crystals can be neglected compared to that of graupel, the rates of change for cloud water ($\Delta_{\text{DRYG}} r_c$) and ice crystals ($\Delta_{\text{DRYG}} r_i$) follow Eq.(13-15), with the following efficiencies:

$$
E_{cg} = 1 \qquad \text{and} \qquad E_{ig} = 0.01 \ e^{0.1(T-T_t)}
\tag{17}
$$

For raindrops and snow/aggregates whose fall speed cannot be neglected compared to the graupel fall speed, the rate of change is detailed in the following.

### 3.1.3 Cases with particles of non-negligible fall speeds

When both $X$ and $Y$ have non-negligible fall speeds, it becomes difficult to solve the integral form of equation (9). Straka and Mansell (2005) approximated the fall speed differences using mean fall velocities, as in Wisner et al. (1972). Seifert et al. (2006) improved the Wisner-like approximation with the notion of characteristic fall velocity difference. Milbrandt and Yau (2005) introduced mass-weighted fall velocities based on Murakami (1990). The numerical technique suggested by Ferrier (1994) has been adopted in LIMA. Following Walko et al. (1995) and as in LIMA v1.0, the numerical solutions of integrals involving the collection kernels are precomputed in the $[\lambda_x^{min}, \lambda_x^{max}]$ range (logarithmic scale in LIMA v2.0) and stored in look-up tables. New tables are also generated specifically for the number concentrations in LIMA v2.0. $\lambda_x^{min}$ and $\lambda_x^{max}$ are set to $10^3$ and $10^7$, respectively, for raindrops, graupel and hail, while for snow/aggregates, they are set to 50 and $5 \times 10^{10}$. Each order of magnitude of $\lambda$ is discretized over 10 points. Equation (9) can be rewritten as:

$$\Delta_{\text{COL}} r_y = \frac{1}{\rho_a} \frac{\pi}{4} \left( \frac{\rho_{00}}{\rho_a} \right)^{0.4} N_x N_y \Lambda_r(\lambda_x, \lambda_y) \Delta v_{xy}(\lambda_x, \lambda_y), \tag{18}$$

where

$$\Delta v_{xy} = a_y \Lambda_r(\lambda_x, \lambda_y)^{-1} \int_0^\infty \left\{ \int_0^\infty E_{xy}(D_x + D_y)^2 |c_x D_x^{d_x} - c_y D_y^{d_y}| D_y^{b_y} g_y(D_y) \, dD_y \right\} n_x(D_x) \, dD_x \tag{19}$$

The normalization factor $\Lambda_r(\lambda_x, \lambda_y)$ is obtained by removing $E_{xy}$ and the absolute fall speed difference in Eq.(19), thus leading to the formal expression:

$$
\begin{aligned}
\Lambda_r(\lambda_x, \lambda_y) \quad &= \int_0^\infty \left\{ \int_0^\infty (D_x + D_y)^2 D_y^{b_y} g_y(D_y) \, dD_y \right\} g_x(D_x) \, dD_x \\
&= M_x(2) M_y(b_y) + 2 M_x(1) M_y(1 + b_y) + M_y(2 + b_y).
\end{aligned}
\tag{20}
$$

An expression similar to $\Delta_{\text{COL}} r_y$ can be used for the number concentration:

$$\Delta_{\text{COL}} N_y = \frac{1}{\rho_a} \frac{\pi}{4} \left( \frac{\rho_{00}}{\rho_a} \right)^{0.4} N_x N_y \Lambda_N(\lambda_x, \lambda_y) \Delta v_{N,xy}(\lambda_x, \lambda_y), \tag{21}$$

where

$$\Delta v_{N,xy} = \Lambda_N(\lambda_x, \lambda_y)^{-1} \int_0^\infty \left\{ \int_0^\infty E_{xy}(D_x + D_y)^2 |c_x D_x^{d_x} - c_y D_y^{d_y}| g_y(D_y) \, dD_y \right\} g_x(D_x) \, dD_x \tag{22}$$

and with:

$$
\begin{aligned}
\Lambda_N(\lambda_x,\lambda_y) \quad &= \int\limits_0^\infty \left\{ \int\limits_0^\infty (D_x + D_y)^2 g_y(D_y)\,dD_y \right\} g_x(D_x)\,dD_x \\
&= M_x(2) + 2M_x(1)M_y(1) + M_y(2).
\end{aligned}
\tag{23}
$$

Since $\Delta v_{xy}$ (resp. $\Delta v_{N,xy}$) is a function only of the local values of $\lambda_x$ and $\lambda_y$, a two-dimensional look-up table is created that contains the numerical solutions of Eq.(19) (resp. Eq. 22) for a series of logarithmically spaced ($\lambda_x$, $\lambda_y$) pairs in the physically expected ranges $[\lambda_x^{min}, \lambda_x^{max}]$ and $[\lambda_y^{min}, \lambda_y^{max}]$, respectively. Then, a bilinear interpolation with respect to the tabulated values of $\lambda_x$ and $\lambda_y$ is used to accurately estimate $\Delta v_{xy}$ (resp. $\Delta v_{N,xy}$).

The following processes are concerned:

1. **Rain accretion on aggregates (ACC)**: As with cloud droplet riming, it is assumed that the collection of small raindrops by an aggregate does not alter its structure, while the collection of larger raindrops transforms an aggregate into a graupel. Based on Ferrier (1994), the mean diameter beyond which raindrop-collecting aggregates are considered graupels ($D_r^{lim}$) is defined by calculating the density of the newly formed aggregate-raindrop mixture ($\rho_{sr}$) from:

$$
\frac{\pi}{6}\rho_w D_r^3 + \frac{\pi}{6}\underbrace{\left[ a_s \frac{6}{\pi} D_s^{b_s-3} \right]}_{\rho_s} D_s^3 = \frac{\pi}{6}\rho_{sr} D_s^3,
\tag{24}
$$

where $\rho_w$ is the liquid water density. If $\rho_{sr} > 0.5(\rho_g + \rho_s)$, the new particle is categorized as a graupel of density $\rho_g$. Since graupels are considered as quasi spheroids ($b_g \sim 3$ in Table 1), $D_r^{lim}$ can be expressed as:

$$
D_r^{lim} = \left[ \frac{3}{\pi} \frac{(a_g - a_s D_s^{b_s-3})}{\rho_w} \right]^{1/3} D_s.
\tag{25}
$$

2. **Graupel dry growth (DRYG)**: Rates involving drops ($\Delta_{\text{DRYG}} r_r$ in Eq.(16)) and snow/aggregate particles ($\Delta_{\text{DRYG}} r_s$ in Eq.(16)) are estimated with different efficiencies that are: $E_{cg} = E_{rg}$ and $E_{ig} = E_{sg}$ as in Ferrier (1994) with revisions in Ferrier et al. (1995).

3. **Graupel and hail wet growth (WETG/WETH)**: In LIMA v2.0, the treatment of the competing dry/wet growth (DRYG/WETG) regimes follows Lin et al. (1983). Hence, these processes are computed by integrals of the form of Eq.(14). Similarly, as for the dry growth mode: the number concentration tendencies of ice crystals, snow and cloud droplets due to graupel wet growth ($\Delta_{\text{WETG}} N_i$, $\Delta_{\text{WETG}} N_s$ and $\Delta_{\text{WETG}} N_c$, respectively) are estimated using Eq.(15) and (23). The raindrop number concentration rate is given by: $\Delta_{\text{WETG}} N_r = \Delta_{\text{WETG}} r_r \times \frac{N_r}{r_r}$.

The formation rate of the hail particles is derived from the WET and DRY growth modes of the graupel particles. The partial conversion of graupel particles into hailstones is then approximated by:

$$\Delta_{\text{WET}} r_h = \left(\frac{\partial r_g}{\partial t}\right)^* \times \frac{DRY}{DRY + WET} \tag{26}$$

where $(\partial r_g / \partial t)^*$ is the sum of the $r_g$ tendencies before conversion to hail.

Once formed, hail develops exclusively in the wet growth mode. When cloud droplets and hail mixing ratios decrease below a threshold value, hail is converted back into graupel (COHG) following a linear percent conversion rate. The same rate is applied to the number concentration.

4. **Snow self collection (SC)**: During this process, the snow mixing ratio does not change while the snow number concentration decreases. This process is parameterized using a snow-snow collection efficiency similar to Milbrandt and Yau (2005): $E_{ss} = 0.05 e^{0.1(T-T_t)}$.

## 3.2 Source/Sink terms other than collection

1. **Conversion of pristine ice crystal to form aggregates (CNVS)** In LIMA v2.0, the conversion rate follows Harrington et al. (1995) parameterization:

$$\Delta_{\text{CNVS}} N_i = \frac{\Delta_{\text{CNVS}} r_i}{a_i D_t^{b_i}}, \tag{27}$$

where $\Delta_{\text{CNVS}} N_i$ and $\Delta_{\text{CNVS}} r_i$ are the number concentration and mixing ratio tendencies of ice crystals due to ice crystals conversion into snow/aggregates as defined in Vié et al. (2016), and $D_t$ is the threshold diameter from which ice crystals are converted into snow aggregates. Here, this threshold diameter is fixed at $D_t = 125$ μm. The gain of snow/aggregates number concentration rate in LIMA v2.0 is then given by: $\Delta_{\text{CNVS}} N_s = -\Delta_{\text{CNVS}} N_i$.

2. **Ice particle melting (XMLT)**: For graupel melting (GMLT) process, by analogy to the graupel wet growth mode, the water formed on the surface of particles is shed away to form raindrops. It is assumed that all the raindrops formed have a fixed diameter thus the number concentration tendency of graupel associated with melting is:

$$\begin{aligned}
\Delta_{\text{GMLT}} N_g &= -\Delta_{\text{GMLT}} N_r \\
&= \Delta_{\text{GMLT}} r_g \times \frac{\rho_a}{m_{0r}},
\end{aligned} \tag{28}$$

where $m_{0r}$ is the mass of a $0.72$ mm raindrop diameter.

The melting process of snow particles/aggregates (CMEL) is different from that of graupels. Tunnel experiments (Mitra et al., 1990) have shown that, at the onset of melting, water from melting aggregates is trapped in the interstices of

their porous structure. Melting aggregates therefore tend to become denser and to bear some resemblance to mixed-phase graupels. Consequently, it is assumed that a the portion of melting aggregates composed of a mixture of water and aggregates which is dense enough to become a graupel is transferred into (melting) graupels at a rate $\Delta_{\mathrm{CMEL}}N_s$ proportional to $\Delta_{\mathrm{SMLT}}N_s$:

$$\Delta_{\mathrm{CMEL}}N_s = \alpha_{s \to g}\Delta_{\mathrm{SMLT}}N_s. \tag{29}$$

and

$$\Delta_{\mathrm{SMLT}}N_s = \Delta_{\mathrm{SMLT}}r_s \times \frac{N_s}{r_s}, \tag{30}$$

where $\Delta_{\mathrm{CMEL}}N_s$ and $\Delta_{\mathrm{SMLT}}N_s$ are the number concentration tendency of snow due to conversion melting and melting, respectively, and $\Delta_{\mathrm{SMLT}}r_s$ is the mixing ratio tendency of snow due to melting. In this scheme, $\alpha_{s \to g}$ has a value of 2, meaning that an equal portion of solid ice and liquid water is required to build a graupel-like structure during the melting of aggregates.

3. **Sedimentation (SED)**: The rates are computed by integration over the PSDs for the mixing ratios and the number concentrations as in Cohard and Pinty (2000a), and so now fully extended to the ice-phase hydrometeors. The "two-moment" sedimentation rates enable the size sorting during the fall of the particles.

## 4   3D case study of an idealized severe storm

### 4.1   Initial model set-up

A case of supercellular storm (Barthe et al., 2005) has been chosen to illustrate the impact of the mixed-phase full two-moment LIMA-v2.0 scheme on the development and structure of a deep convection cloud. The simulation of the 3D supercellular storm starts from the initial sounding of Klemp and Wilhelmson (1978). Convection is initiated by a warm bubble of 1.5 K and 10 km radius which is located in the planetary boundary layer, in the center of the domain. The simulation lasts 90 minutes. It is performed over a domain of $200 \times 200 \times 60$ gridpoints with a grid spacing of 500 m on the horizontal and 250 m on the vertical. Horizontal and vertical velocities are advected with a fourth-order scheme centered on space and time associated with a leap-frog temporal scheme. The Piecewise Parabolic Method (Colella and Woodward, 1984) associated with a forward-in-time temporal scheme is used to advect meteorological and scalar variables. The time step is 3 s. A 3D turbulence scheme (Cuxart et al., 2000) is activated.

The Meso-NH model is used to examine the differences between a simulation using the improved and full 2-moment microphysics scheme (hereinafter referred to as LIMA2), and another one using the original partially 2-moment microphysics scheme (hereinafter referred to as LIMA1). The LIMA1 and LIMA2 simulations are fed with a superposition of several aerosol

modes. Aerosols number concentrations are state variables in the model. Each aerosol mode is defined by distinct nucleation properties, and can be used as CCN to form cloud droplets (following Cohard et al., 1998) or as IFN to form ice crystals (following Phillips et al., 2013). A single CCN mode and a single IFN mode are prescribed in this study. The CCN concentration is constant between the ground and 1000 m altitude and set to $300 \times 10^6$ m$^{-3}$. Then it decreases exponentially up to 10,000 m where it reaches a constant value of $10 \times 10^3$ m$^{-3}$. The IFN concentration is homogeneous over the vertical ; it is set to 1000 L$^{-1}$.

## 4.2 Modeled storm evolution

First of all, the location of the maximum of precipitation is the same in both simulation (X = 46.5 km, Y = 49 km in Fig. 2). The area covered by the accumulated precipitation at the ground is quite similar (315 km$^2$ in LIMA1 and 395 km$^2$ in LIMA2) in the two simulations. However, the maximum value of the accumulated precipitation is larger in LIMA2 (122 mm) than in LIMA1 (75 mm). Concerning hail, LIMA1 simulates a much larger area of accumulated hail precipitation than LIMA2 (98 vs 53 km$^2$). The LIMA1 simulation shows that hail is spread over almost the same area as rain. In contrast, the LIMA2 simulation produces more localized hail precipitation, close to the maximum accumulated precipitation. The LIMA2 simulation produces a dramatically reduced accumulated hail precipitation at the ground compared to the LIMA1 simulation (3 mm in LIMA2 vs. 15 mm in LIMA1). Although it is not possible to draw conclusions about the amount of hail reaching the ground, the rather restricted location of hail at the ground with the LIMA2 simulation is more in line with the supercell patterns shown in Doswell III and Burgess (1993) and Kumjian and Ryzhkov (2008).

The time evolution of the mean updraft and downdraft (averaged vertical wind speed from 0 to 8 km) shown in Fig. 3 is used to compare the dynamics of the simulated storms and its relation to the production of precipitation. The mean downdraft and updraft in the two simulations are very close up to 32 min. After 32 minutes, however, the differences between the two simulations are more pronounced. The mean updraft between 32 and 50 min is up to 1.5 m s$^{-1}$ higher in LIMA2 compared to LIMA1. The downdraft reaches its mean maximum at the time when the hail precipitation reaches its maximum (40 min for LIMA1 as compared to 50 min for LIMA2). For the mean precipitation rates, the differences are significant. LIMA2 produces a clear peak around 34 mm h$^{-1}$ between 26 and 28 minutes, while LIMA1 produces a peak around 25 mm h$^{-1}$ between 22 and 34 minutes. The average precipitation rate then decreases in the two simulations and the two curves almost merge from 40 minutes onward. From 50 minutes, the mean rainfall in the two simulations is low, in the order of a few mm h$^{-1}$. The most significant differences are in the average hail precipitation rates. In LIMA1, hail particles start to reach the ground at 26 min and the hail rate increases very rapidly, reaching 8 mm h$^{-1}$ after 44 minutes. Significant oscillations in the hail precipitation rate are observed in this simulation, with maxima at 30 min (3.5 mm h$^{-1}$), 34 min (4.5 mm h$^{-1}$), 44 min (7.5 mm h$^{-1}$) and 56 min (6 mm h$^{-1}$). On the other hand, in the LIMA2 simulation, the hail precipitation rate starts at the same time, but increases much more slightly, reaching a peak of 2 mm h$^{-1}$ at 48 min. After 70 min, there is virtually no hail reaching the ground in LIMA2. Thus, the precipitation rates indicate a significant difference in the microphysical state of the storm when using LIMA1 with a single moment parameterization of the snow aggregates, graupel or hail versus the use of a 2-moment

parameterization of these species, while the dynamics of the storm does not differ significantly between the two simulations.

We now examine the results of the two simulations at two different times. The first time selected is 26 min. At this time, the maximum rainfall is about to come while hail yet reaches the ground, and the major dynamical features of the two modeled systems are quite similar. The second selected time is 42 minutes, before the maximum of hail precipitation, and when the mean updraft reaches a threshold value and the subsidence becomes more intense.

Figure 4 compares some features of the two simulations at the selected times (vertical lines Fig. 3). After 26 minutes of simulation (Fig. 4a and c), surface dynamics (red and blue lines) and height dynamics (orange and cyan lines) are similar, although there are slight differences in the location and extent of subsidence. The differences become more pronounced after 42 minutes of simulation. Rainfall is more intense in LIMA2 (by a factor of about 2) with a significant reduction of hail (by a factor of about 10). Again, while uplift zones are similar over the first 4 km and between 4 and 8 km, subsidence zones are different. In the LIMA2 simulation, the subsidence zones are located upstream (with respect to the surface currents) over the first 4 km, whereas in LIMA1 they are centred on the hail precipitation. The cloud composition is detailed in Fig. 5 along the oblique axis.

Figure 5 summarizes the vertical distribution of all hydrometeors through the maximum updraft. Each pie chart shows how the total water content (shaded areas) is distributed among all categories of hydrometeor. As shown in Fig. 3, the maximum simulated vertical velocity is similar in both simulations after 26 minutes. In the two simulations, the vertical and horizontal extension of the system is similar at this time of the simulation. However, there are marked differences observed in the cloud and rain liquid water mixing ratios occurring at sub-zero temperatures. At 26 min, cloud droplets (orange color in the pie charts) reach 7 km altitude in LIMA2 while they do not rise above 6 km in LIMA1. The same behavior is observed for raindrops (blue color in the pie charts); they reach 6 km altitude in the convective region of the LIMA2 simulation while they are found 2 km below in LIMA1. The proportion of raindrops mass above the $0°C$ isotherm is much higher in the LIMA2 simulation, with proportions exceeding 75% at some grid points. In contrast, they do not exceed 25% around the $0°C$ isotherm in the LIMA1 simulation. At 26 min, more than 50% of the total water mass at the top of the system consists of ice crystals (black color in the pie charts) in LIMA2, whereas it is mainly snow (light blue color in the pie charts) and graupel (yellow color in the pie charts) in LIMA1. Hail (pink color in the pie charts) is found at lower altitudes in LIMA1 compared to LIMA2, and reaches the ground only in LIMA1.

After 42 minutes of simulation (Fig. 5b and d), the vertical extent of the system is slightly larger in LIMA2 (by about 1 km). As also shown in Fig. 5 a and c, LIMA2 produces higher liquid water content at higher levels, and the top of the cloud is mainly composed of ice crystals in LIMA2 compared to snow in LIMA1. Snow appears to be much more localised outside the lift zones in LIMA2, whereas it is present at almost all grid points above the $0°C$ isotherm in LIMA1. Although hail precipitation at the ground extends over 5 km in both simulations, the proportion and mass of hail at the ground are lower in LIMA2. Thus, a larger fraction of the simulated hail remains in the cloud in LIMA2.

The temporal evolution of the horizontal-mean mixing ratio of each vertical profile of hydrometeors for which the thickness of the precipitating hydrometeor is greater than 0.5 mm is shown in Fig. 6. This threshold has been chosen in order to cover all the cloudy areas of the simulation domain. The corresponding temporal evolution of the vertical profiles of number concentrations are shown in Fig. 7. The $C\lambda^x$ relationship is used to diagnose the snow, graupel and hail number concentration shown for the LIMA1 simulation. Since the modifications of the LIMA scheme concern mostly the ice phase, we comment on the differences between the two simulations in the ice phase. The differences in the liquid phase are only consequences of what is observed in the ice phase.

The first panel in Fig. 6 (a and g) shows the mass mixing ratio of the ice crystals. During the early stage of the system, ice crystal growth is similar in both simulations but the ice crystals mixing ratio is significantly higher in LIMA2 compared to LIMA1. Ice crystal mixing ratio reaches 0.01 $\mathrm{g\,kg^{-1}}$ after 18 minutes at a height of about 5 km in the LIMA1 simulation. In the LIMA2 simulation, the mixing ratio reaches up to 0.35 $\mathrm{g\,kg^{-1}}$ after 20 minutes and reaches a second significant peak of 0.3 $\mathrm{g\,kg^{-1}}$ between 32 and 38 minutes around 9 km height. Although the number concentrations are similar for the first 30 minutes in LIMA1 and LIMA2, then the number mixing ratios are significantly higher in the LIMA2 simulation (Fig. 7 a and g). Ice crystals mass accumulates between 4 and 6 km up to 20 minutes in LIMA2, while in LIMA1 it is consumed more efficiently. After about 30 minutes of the simulation, ice crystals accumulate between 8 and 10 km, both in terms of mixing ratio and number concentration. But once again, this accumulation is much larger when all the hydrometeors are represented using two moments (LIMA v2.0) .

Figure 5 shows that in LIMA2 the snow was located below ice crystal accumulation, mainly in the stratiform part at the end of the simulation. These successive layers inside convective systems are also observed by Barnes and Houze Jr. (2014). The vertical profiles of Fig. 6 (b and h) confirm this observation. Snow/aggregates mixing ratio is present at an altitude of 6 km in LIMA2 compared to 4 km in the LIMA1 simulation. The maximum of 0.7 $\mathrm{g\,kg^{-1}}$ is reached after 20 minutes in the LIMA1 simulation and then decreases rapidly (after 40 minutes) to about 0.2 $\mathrm{g\,kg^{-1}}$ by the end of the simulation. In LIMA2 it is about 0.2 $\mathrm{g\,kg^{-1}}$ after 26 minutes of simulation and this value increases only up to 0.4 $\mathrm{g\,kg^{-1}}$ until 50 minutes. The snow/aggregates are spread between 3 and 10 km height in the LIMA1 simulation, but it is more concentrated between 4 and 8 km in LIMA2. In the corresponding mean number concentration profiles (Fig. 6 b and d), the differences are much more pronounced due to the empirical relationship to estimate the number concentration used in the one-moment version of the code (more explanation below). Since snow/aggregates are initiated by autoconversion of pristine ice and grow by aggregation of pristine ice or self-collection (see Fig. 1 and Table 2), it is expected that the number concentration of snow is less than (but close to) the number concentration of ice crystals. Indeed, in LIMA2, $N_i$ is about one order of magnitude higher than $N_s$. However, in LIMA1, $N_i$ is five orders of magnitude higher than $N_s$ suggesting that the diagnostics of $N_s$ are not appropriate.

Concerning graupel (Fig. 6 c and i) and hail (Fig. 6 d and j), the mean mixing ratio profiles are similar with respect to the initiation time and the vertical extent. However, the mixing ratios are much lower for graupel and higher for hail in LIMA2 compared to LIMA1. For these two species, the associated number concentration profiles are lower compared to the diagnostic used in LIMA1. Between 30 and 40 minutes and between 6 and 8 km, hail profiles are most distinct. As shown in Fig. 1 and Table 2, graupel is mainly produced by the freezing of rain drops, the riming of snow or the accretion of rain and aggregates.

Therefore, $N_g$ should be equal to or smaller than $N_s$ or $N_r$. This is the case in the LIMA2 simulation. However, Fig. 7 clearly shows that $N_g$ is significantly larger than $N_s$ and $N_r$ in the LIMA1 simulation.

Noticeable differences in cloud droplet vertical profiles are simulated between 20 and 40 minutes. Even if the maximum values are similar, the vertical extend of mixing ratio exceeding $0.15\,\mathrm{g\,kg^{-1}}$ is higher in LIMA2 than in LIMA1. Whereas, after 20 minutes of simulation the main effect on the rain drops can be seen as more drops are simulated between 4 and 6 $\mathrm{km}$ in LIMA2.

    The mean vertical profiles of the conversion rates in Fig. 8 provide a further explanation of the differences between the
465 LIMA1 and LIMA2 simulations discussed above. The profiles represent the average transfers weighted by the mixing ratio when the system is initiated between 14 and 24 minutes of simulation. This figure shows the efficiency of the different non-collection processes as in Fig. 1a (Fig. 8a-d) and the average transfers for the processes involving collection (Fig. 8e-h). At system initiation, the non-collection processes involving ice crystals (Fig. 8a) are similar for both simulations. We zoomed in on the process of converting ice crystals into snow (CNVS), which is the process that initiates the snow. The differences in
snow mixing ratios highlighted in the previous figures are therefore linked to the growth processes. Figure 8e shows that the aggregation of ice crystals on snow is less effective when the concentration is prognostic (LIMA2). The processes involving snow and graupel are much less effective in the LIMA2 simulation. This is true for all processes (collection and non-collection).

    There are several explanations for the differences between the LIMA1 and LIMA2 simulations. The first one is related to
475 the mean different sizes of the snow, graupel and hail particles when the number concentration of the precipitating ice particles is prognostic. It can be seen from Fig. 9 that for these three species and for a given mixing ratio, the number concentration can vary by several orders of magnitude. In general, the mean size of snow and graupel is smaller for low contents in LIMA1 than in LIMA2. This difference in the mean size of the ice precipitating species leads to changes in the efficiency of the deposition and collection processes. Another explanation for the different levels at which the maximum concentrations are simulated as shown
in Fig. 5, 6 and 7 can come from sedimentation, which takes into account the dimensions of the hydrometeors in LIMA2. For hail, for example, we can see in Fig. 8d that sedimentation occurs in similar proportions for the two simulations, but that hail is carried down to the ground more quickly. This explains why there seems to be more hail on average within the system but less precipitation in LIMA2. Subsequently, this hail, which will be represented under the $0°\,\mathrm{C}$ isoline, will have more time to melt and will increase rainfall.

    We compare our results from LIMA1 and LIMA2 simulations to airborne observations from the HyMeX (Hydrological Cycle in the Mediterranean Experiment) field campaign (Ducrocq et al., 2014). Taufour et al. (2018) performed a linear regression on the mixing ratio vs. number concentration data from the observed PSD (black line in Fig. 9). When comparing the airborne data of number concentrations to the mass mixing ratios of snow and graupel, the LIMA v2.0 scheme can reproduce
well the variability of $N - r$ occurrences. Indeed, at very low mixing ratios, very high number concentrations are assumed in the parameterization of the snow concentration in the 1-moment version of the scheme (Fig. 9a). This is confirmed by the evolution of the maximum diagnosed number concentration (Fig. 7) which is anti-correlated with the maximum snow mixing

ratio. This actually resolves a known issue of single-moment scheme. A better estimate of the optical and radiative properties of the cloud using the full 2-moment version of the LIMA scheme is thus expected.

Note that using LIMA v2.0 instead of LIMA v1.0 in this 3D case study results in a 17 % increase in computation time. This is mainly explained by an increase in the numerical cost for the advection of meteorological and scalar variables (+ 42 %), the on-line budgets calculation (+ 16 %), turbulence (+ 12 %), and the microphysical scheme (+ 10 %). As a reference, advection of meteorological and scalar variables, turbulence, microphysics and on-line budgets are responsible of 13.8, 8.7, 10.6 and 31.0 % of the total computation time, respectively, in the LIMA2 simulation.

## 5    Summary and Conclusions

A new full two-moment microphysics scheme, LIMA v2.0, has been developed from the previous partially two-moment scheme LIMA implemented in the Meso-NH cloud-resolving model. The new scheme now integrates a set of equations of the number concentrations tendencies in addition to those of the corresponding mass mixing ratios. Six hydrometeor categories are considered with new features concerning especially the precipitating ice phase categories including snow-aggregates, graupel and hail. Consequently, 18 additional conversion rates, specific to the number concentration equations, are introduced on the basis of the particle size distributions.

The LIMA v2.0 scheme is numerically stable and more physically based than LIMA. It is tested in the host model Meso-NH for an idealized deep convection case. The results show that LIMA v2.0 works satisfactorily and so brings an original support to the study of convective cloud structures up to the development of hailstorms. The inclusion of prognostic number concentrations of snow, graupel, and hail enables more degrees of freedom to represent the size distribution of these hydrometeors. Thus, it expands the possible microphysical states of deep convective clouds, in which the ice phase is present, with a significant impact on the precipitation fields, the microphysical structure and the dynamical evolution of the clouds.

Generally speaking, adding number concentration as a second prognostic moment to calculate size-dependent microphysical conversions leads to an increase of the mass mixing ratio of small ice crystals inside clouds and a decrease of snow and graupel hydrometeors in comparison to the results produced by the former LIMA scheme. In-depth analyses of the microphysical processes affected by the prognostic number concentrations of snow, graupel, and hailstone suggest different feedbacks between rain and ice phase hydrometeors when compared to the results obtained with a single-moment parameterization of snow, graupel and hail. Applying the full 2-moment microphysical scheme not only modifies the mass mixing ratio of these hydrometeors but also impacts on the mass mixing ratios of other hydrometeors and the altitudes where the variety of hydrometeors is maximized. Firstly, the consideration of prognostic number concentration in LIMA v2.0 initially slows down the formation of snow and graupel in the tested deep convective case. Ice crystals, cloud water droplets and rain therefore accumulate between 4 and 6 km, whereas these hydrometeors were consumed by snow and graupel in LIMA v1.0. The use of the fixed parameters $C$ and $x$ to crudely estimate the concentration of snow and graupel in version 1 of the scheme is the main reason for this. Secondly, in LIMA v2.0, the use of 2 moments for snow creates a real distinction between the ice crystals accumulated in the upper part of the cloud (around 9 to 10 km) and the snow at the lower levels. Finally, it introduces a better opportunity for both raindrops

and hailstones to reach the ground earlier or to reinforce locally the precipitation. The intensification of the rain in LIMA v2.0 is connected to the melting of the hail and graupel. Therefore the hail precipitation amount is lower in LIMA v2.0.

It is believed that increasing the degrees of freedom to represent the hydrometeors size distribution is the key to improve the skills of microphysics schemes. The snow aggregate particles seem to be the most affected hydrometeor. The single moment
parameterization of snow in LIMA v1.0 is already known to lead to discrepancies with observations (Taufour et al., 2018; Wurtz et al., 2021). Wurtz et al. (2023) developed a new diagnostic of the $\lambda$, leading to improvements in the diagnostic number concentration and associated processes.

Certain further evaluations of LIMA v2.0 are still needed, e.g., conducted using well-documented observed cases under different environmental conditions. The two-moment parameterization impacts the simulated radar reflectivities so that greater
variability in hydrometeor mean sizes leads to greater variability in radar reflectivity. Even if it is hard and a little bit tedious to compare the computed reflectivities to mean observed vertical profiles, the results presented in this study evidence the need to check more precisely the contribution of the different types of hydrometeors. It is expected that exploiting multiple parameters (Doppler shift and polarization diversity) of radar measurements (Caumont et al., 2006; Brown et al., 2017), is the best way to quantitatively assess the accuracy of microphysics schemes regarding the location of the different hydrometeor types. The
polarimetric radar operator described by Augros et al. (2016), will soon be adapted to the new scheme for a further evaluation of supercell characteristics.

LIMA enables the interactions of aerosols (chemical composition and size distribution) with cloud microphysics (Vié et al., 2016; Hoarau et al., 2018a). Therefore, this new version of the scheme provides a solid basis to study of the influence of atmospheric aerosols on the size of hailstones that reach the ground, which can be a major cause of damage to agriculture
and property. Further studies following this direction with more carefully designated locations and timing for IFN entering the cloud can address the issues associated with the increase of aerosols due to a polluted environment or more specifically a targeted cloud seeding experiment. Nevertheless, more evaluations should be conducted, ideally with realistic cases that we are currently working on.

*Code availability.*  The LIMA v2.0 scheme has been implemented in the 5.6 version of the Meso-NH code. This reference version is under
the CeCILL-C license agreement and freely available at http://mesonh.aero.obs-mip.fr/mesonh56 (last access: 20 March 2024). The complete code of Meso-NH v5-6-0 including LIMA v2.0 is available at https://zenodo.org/doi/10.5281/zenodo.11393717. This repository also contains namelists to run the idealized storm and python scripts to reproduce the figures of this manuscript.

**Appendix A:  List of symbols**

*Author contributions.*  M. T. coded the LIMA (v2.0) scheme with B. V., performed the simulations, prepared the figures and initiated the
manuscript. J.-P. P. advised the development of the LIMA (v2.0) scheme through discussions wih M. T. and C. B., and finalized a first

version of the manuscript. C. W. raised the idea to develop the advanced LIMA (v2.0) scheme from the existing LIMA code in Meso-NH. C. B., B. V. and C. W. corrected in deep the manuscript. C. B. led the review process with contributions from J.-P. P. and C. W.

*Competing interests.* The contact author has declared that none of the authors has any competing interests.

*Acknowledgements.* This study was supported by Agence Nationale de la Recherche (ANR) of France under the "Programme d'Investissements
d'Avenir" (ANR-18-MPGA-003 EUROACE), France. This work was also supported by ANR ICCARE under grant ANR-21-CE01-0006. Computations were performed on the local computer cluster of Laboratoire d'Aérologie. J.-P. Pinty wishes to acknowledge CALMIP (CALcul en MIdi-Pyrénées) of the University of Toulouse for access to the "Olympe" supercomputer where useful additional simulations could be performed. The scientific colour maps nuuk, roma and managua (Crameri, 2018) are used in this study to prevent visual distortion of the data and exclusion of readers with colour-vision deficiencies (Crameri et al., 2020).

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

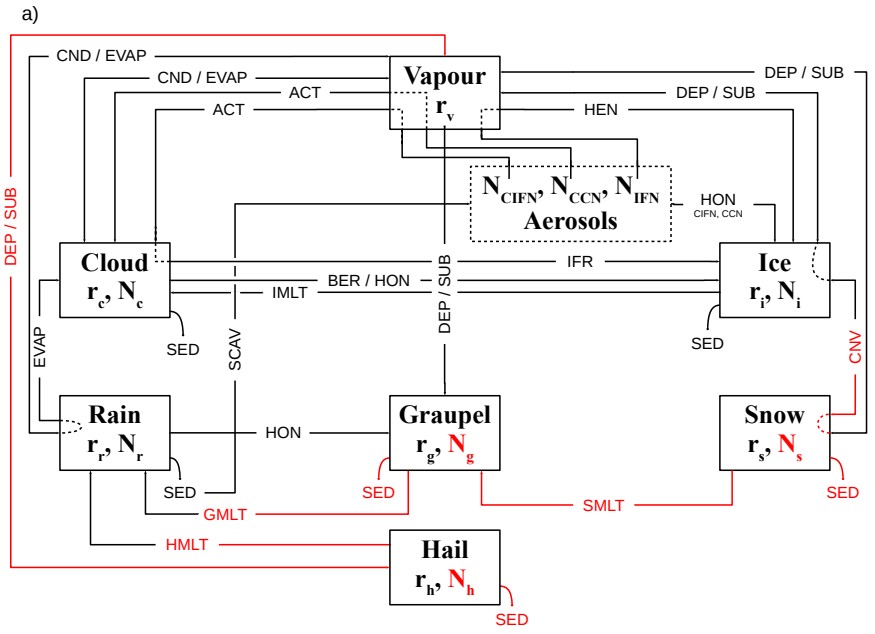

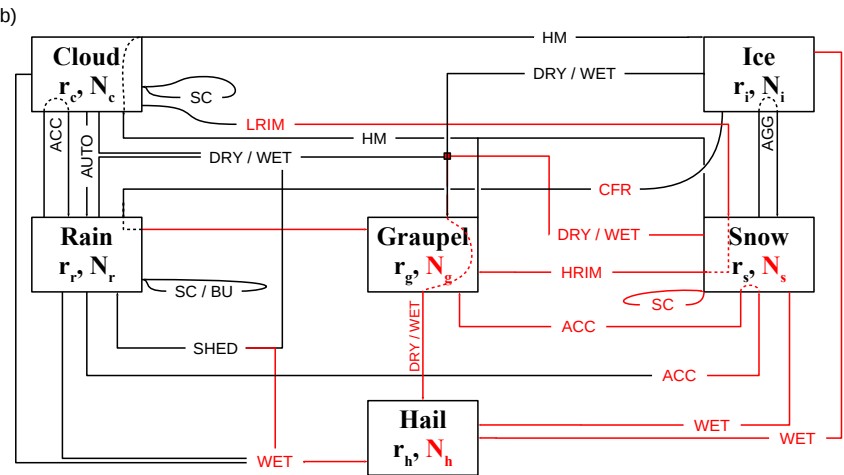

**Figure 1.** Diagrams of the microphysical processes of LIMA v2.0 representing all the processes except collection (a) and all the collection processes (b). Red arrows represent new or modified processes in v2.0, black arrows are identical processes in LIMA v1.0 and v2.0. Prognostic variables for all the hydrometeor species are written in the boxes, with $r_x$ and $N_x$ the mixing ratio and number concentration of the species $x$, respectively. The process label are explained in Table 2.

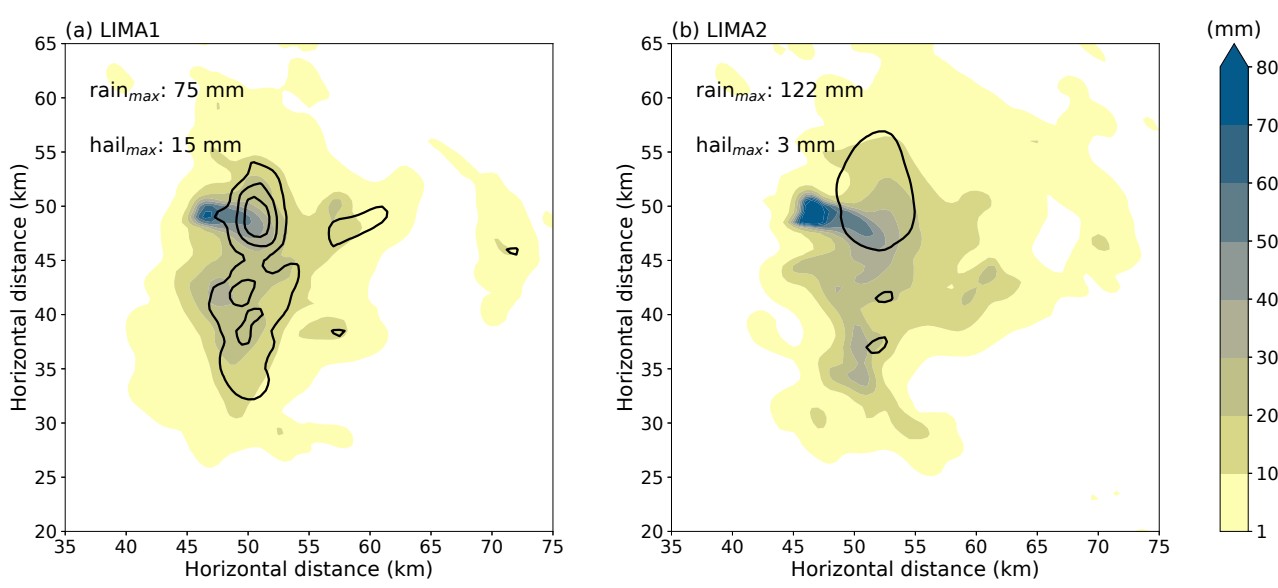

**Figure 2.** Accumulated precipitation (mm) from rain (colors) and hail (black contours at 1, 5 and 10 mm) at the ground for the (a) LIMA1 and (b) LIMA2 simulations.

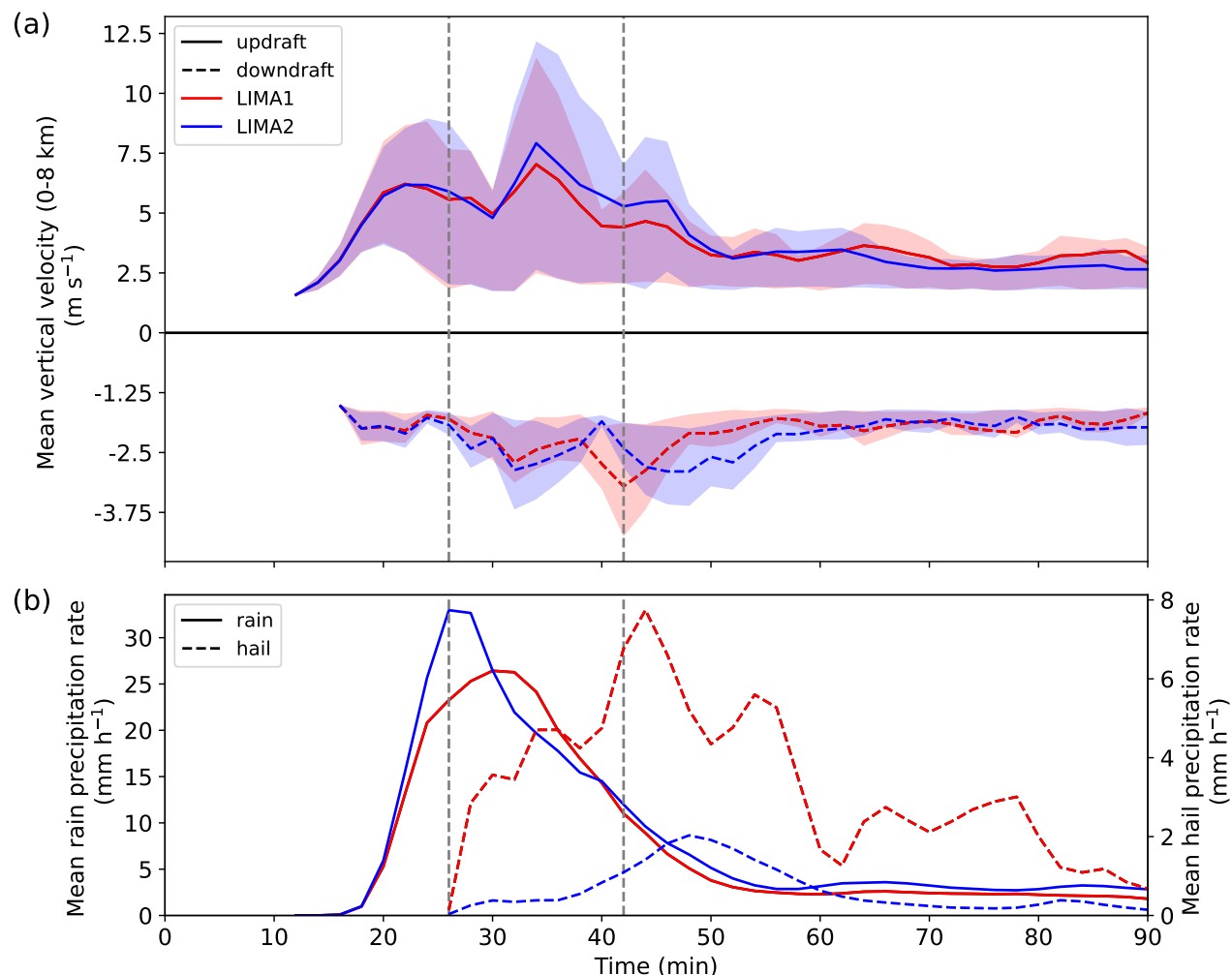

**Figure 3.** Temporal evolution of vertical motions (a) and precipitation rates at the ground (b). In (a), the domain mean vertical updraft speed between the surface and 8 km altitude (top) is represented by the solid lines, and the mean downdraft (bottom) is plotted with dashed lines. The interquartile range is represented in shading. In (b), the mean rain rate of rainy area at the ground is plotted with solid lines and the mean hail rate is plotted with dashed lines. The vertical dashed lines correspond to the time when sections in Fig. 5 are shown.

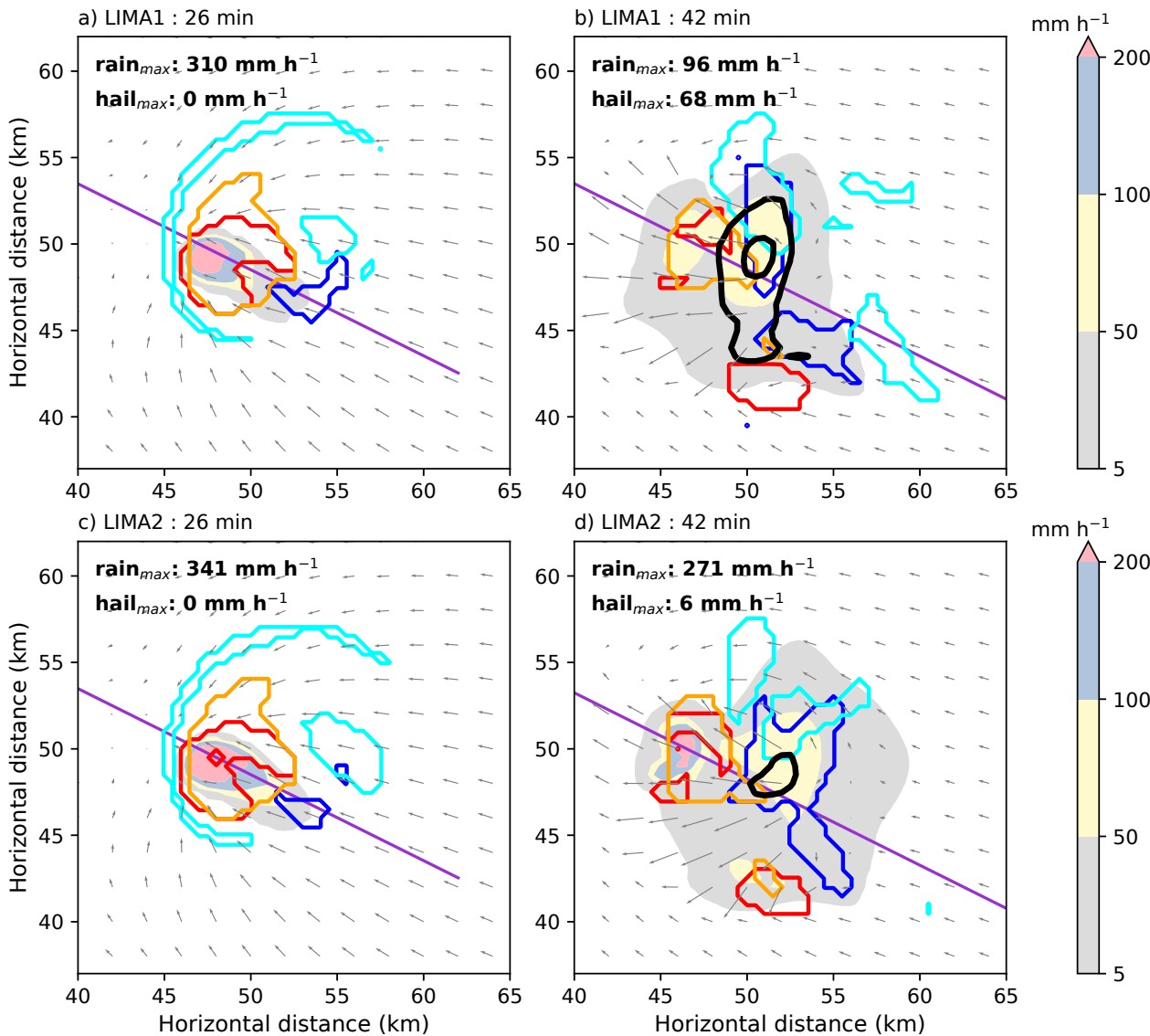

**Figure 4.** Instantaneous rain precipitation (shading) and hail precipitation (black contours at 5 and 50 $\mathrm{mm\,h^{-1}}$) at the ground for the LIMA1 (a-b) and LIMA2 (c-d) simulations at 26 min and 42 min (left and right columns, respectively). The areas where the mean vertical wind speed between 0 and 4 km altitude and between 4 and 8 km altitude exceeds 5 $\mathrm{m\,s^{-1}}$ are plotted with red and orange lines, respectively. The areas where the mean vertical wind speed between 0 and 4 km altitude and between 4 and 8 km altitude is less than -1.5 $\mathrm{m\,s^{-1}}$ are plotted with blue and cyan contours. Grey arrows represent the horizontal wind at 500 m height.

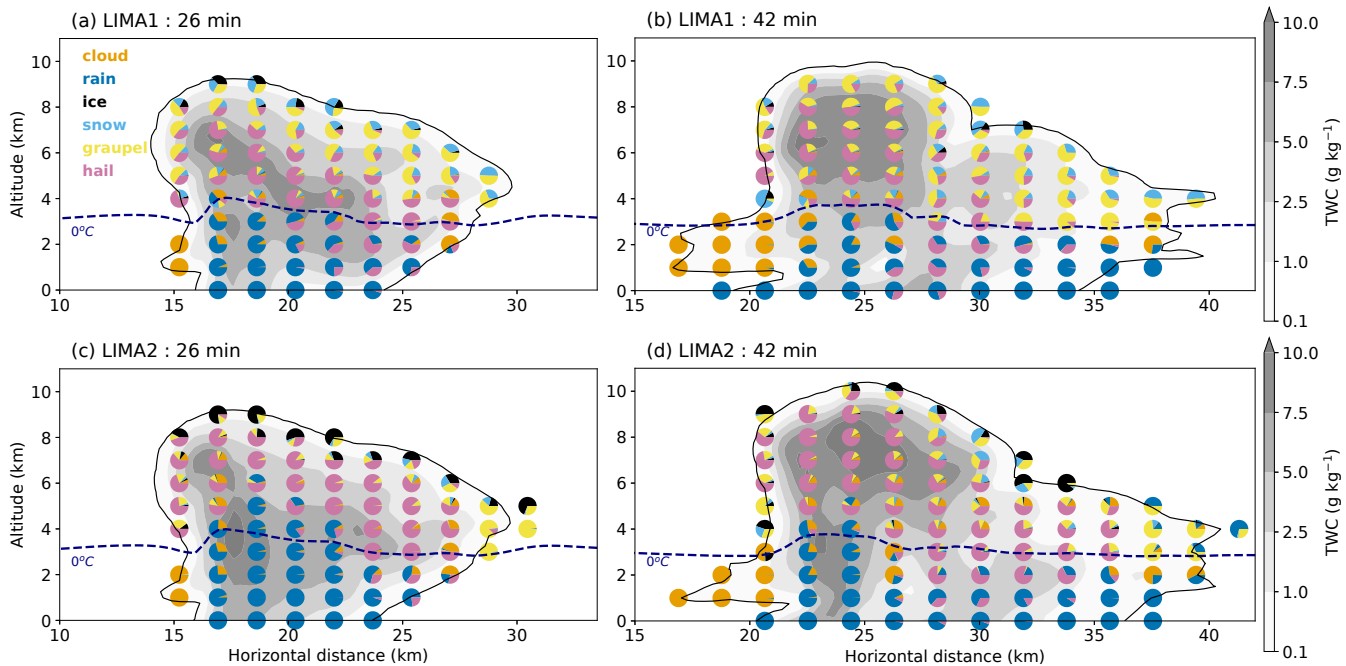

**Figure 5.** Vertical cross sections of the cloud structure passing through the maximum updraft speed and along the zonal axis in LIMA1 (a-b) and LIMA2 (c-d) simulations after 26 min (left) and after 42 min (right) of simulation. The pie charts show the ratio of each hydrometeor mass in the grid mesh (blue for rain, orange for cloud droplets, black for ice, light blue for snow, yellow for graupel, and pink for hail). The gray shaded areas represent the total hydrometeor mass ($\mathrm{g\,kg^{-1}}$) in each grid mesh. The $0°$C (navy blue dashed line) isotherm and the contours of the total water content ($0.1\,\mathrm{g\,kg^{-1}}$, black line) are shown.

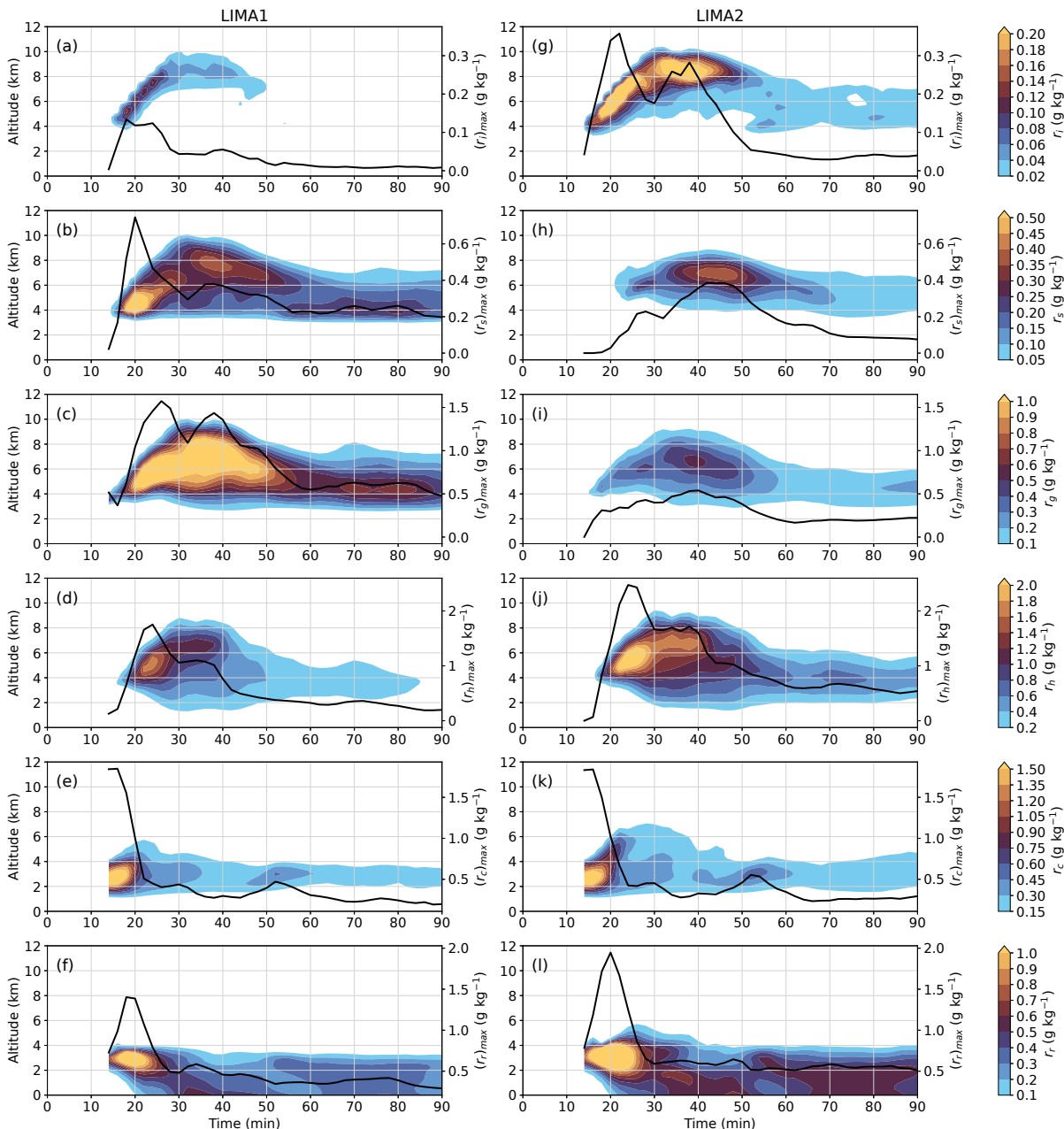

**Figure 6.** Temporal evolution of different hydrometeors vertical mean profiles in LIMA1 (a-f) and LIMA2 (g-l) simulations. The mean mixing ratio (in $\text{g kg}^{-1}$) is computed using columns where the precipitating hydrometeor is greater than 0.5 mm, and is represented, from the top to the bottom, for pristine ice crystals ($r_i$), snow aggregates ($r_s$), graupel ($r_g$), hail ($r_h$), cloud droplets ($r_c$), and raindrops ($r_r$). For each hydrometeor, the time evolution of the maximum mixing ratio is represented by solid lines (right axis).

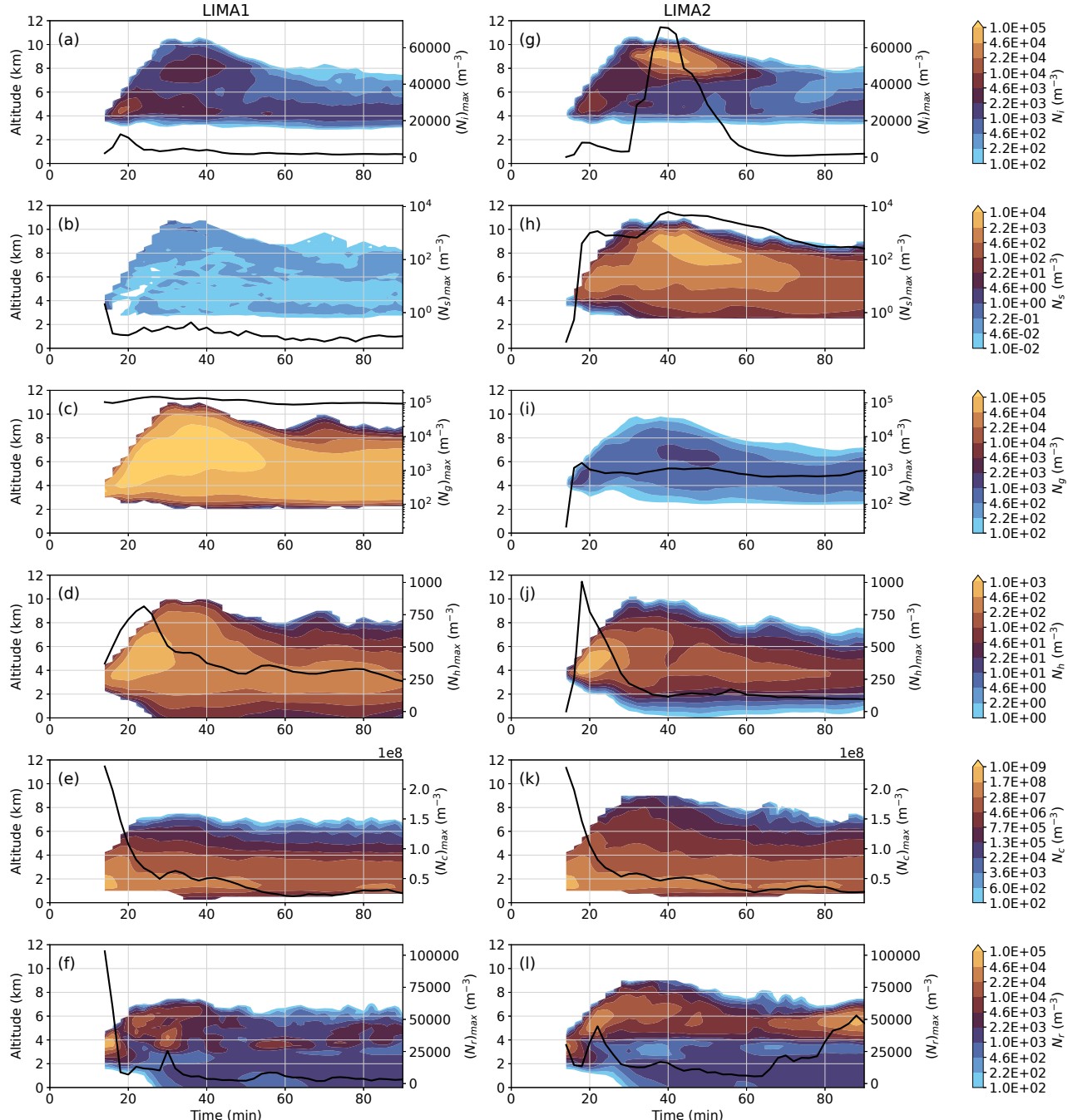

**Figure 7.** Same as Fig. 6 with hydrometeor number concentrations.

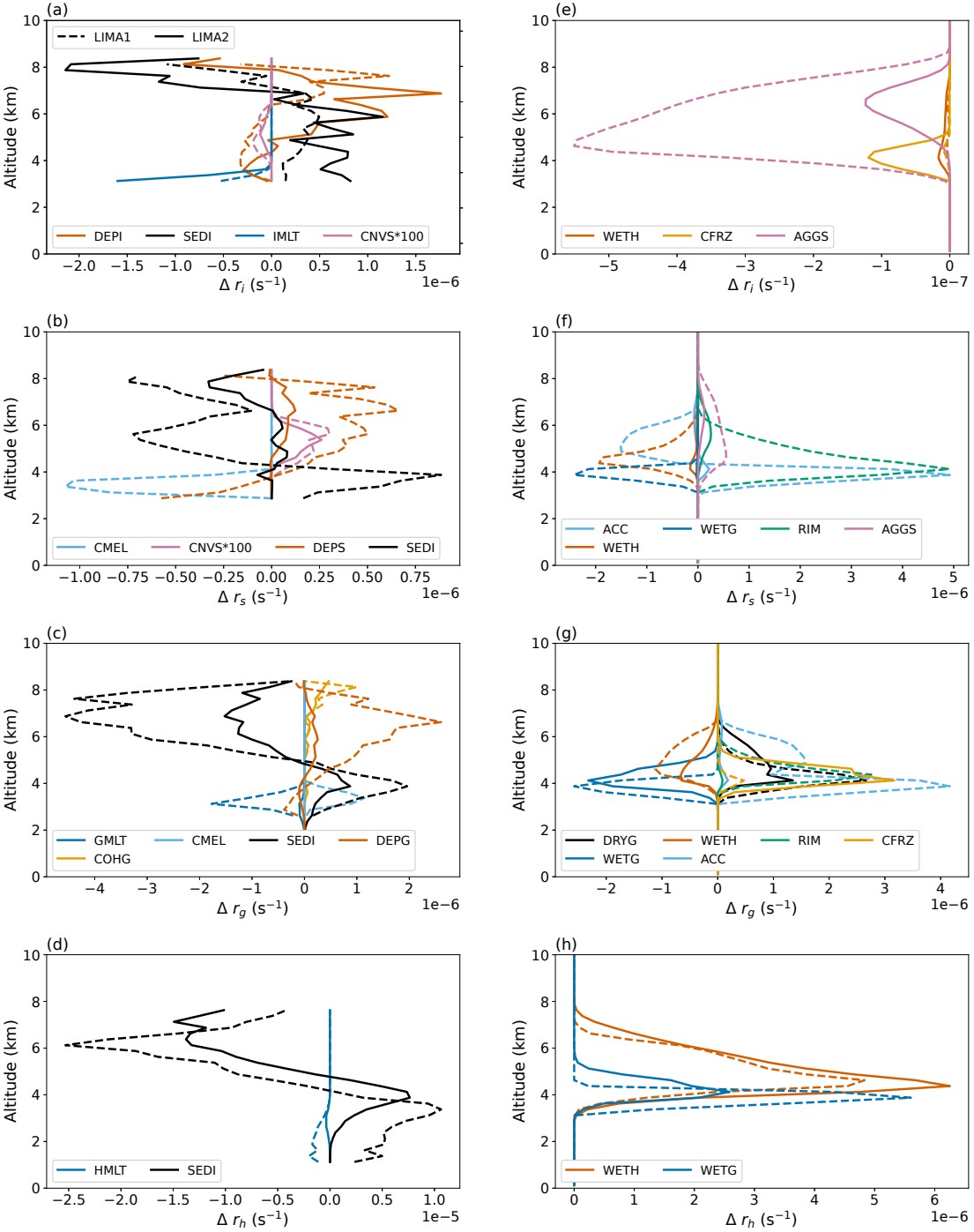

**Figure 8.** Weighted averaged vertical profiles for each process transfer rate (colors) for LIMA1 (dashed lines) and LIMA2 (plain lines) simulations. The profiles are average weighted by the mean mixing ratio of the corresponding hydrometeor (calculated by altitude level for columns where precipitation is above 0.5 mm) between 14 and 24 minutes for the processes except collection (a-d) and all the collection processes (e-h). Only processes with significant transfer rate are plotted. The RIM process corresponds to the sum of the LRIM and HRIM processes in Table 2.

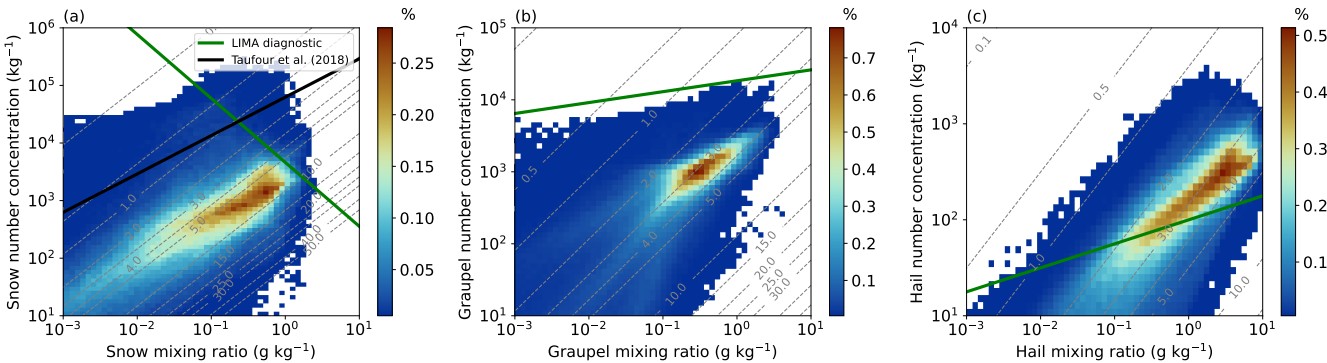

**Figure 9.** Frequency diagram normalized by the total number of points as a function of mixing ratio ($\text{g kg}^{-1}$, x-axis) and number concentration ($\text{kg}^{-1}$, y-axis) of (a) snow/aggregates, (b) graupel, and (c) hail in the LIMA2 simulation. The green lines show the parameterization used in the LIMA v1.0 scheme ($N = C\lambda^x$) and the black line on (a) shows the relationship found using airborne in-situ measurements after Taufour et al. (2018). The dashed grey lines represent the mean volume diameter isolines (in $\text{mm}$).

**Table 1.** Values of the various parameters used to characterize each water category.

| Parameters (unit) | ice | snow | graupel | hail | rain | cloud |
|---|---|---|---|---|---|---|
| $\alpha$ (-) | 3 | 1 | 1 | 1 | 1 | 3 on sea; 1 on land |
| $\nu$ (-) | 3 | 1 | 1 | 8 | 1 | 1 on sea; 3 on land |
| $a$ $(\mathrm{kg\,m^{-b}})$ | 0.82 | 0.02 | 19.6 | 470 | 524 | 524 |
| $b$ (-) | 2.5 | 1.9 | 2.8 | 3 | 3 | 3 |
| $c$ $(\mathrm{m^{1-d}\,s^{-1}})$ | 800 | 5.1 | 124 | 207 | 842 | $3.2\ 10^7$ |
| $d$ (-) | 1.00 | 0.27 | 0.66 | 0.64 | 0.8 | 2 |
| $C$ (-) | | 5 | $5\ 10^5$ | $4\ 10^4$ | $8\ 10^6$ | |
| $x$ (-) | | 1 | -0.5 | -1 | -1 | |
| $\overline{f}_0$ (-) | 1.00 | 0.86 | 0.86 | 0.86 | 1.00 | |
| $\overline{f}_1$ (-) | | 0.28 | 0.28 | 0.28 | 0.26 | |
| $\overline{f}_2$ (-) | 0.14 | | | 0 | | |
| $\mathcal{C}_1$ (-) | $1/\pi$ | $1/\pi$ | 0.5 | 0.5 | 0.5 | |

**Table 2.** List of the LIMA v2.0 microphysical processes.

| Process | Symbol | Sink | Source |
|---|---|---|---|
| warm phase processes | | | |
| heterogeneous nucleation | HENU | $r_v, N_{CCN,CIN}$ | $r_c, N_c$ |
| autoconversion of cloud droplets | AUTO | $r_c, N_c$ | $r_r$ |
| accretion of cloud droplets by raindrops | ACC | $r_c, N_c$ | $r_r$ |
| self collection of droplets | SC | $N_c$ | $\oslash$ |
| self collection of drops / break-up | SC / BU | $N_r$ | $\oslash$ |
| cloud droplets evaporation(/condensation) | EVAP (/ CND) | $r_c, N_c (/r_v)$ | $r_v (/r_c)$ |
| rain drops evaporation | EVAP | $r_r, N_r$ | $N_c, r_v$ |
| sedimentation | SEDI | $r_c, r_r, N_c, N_r$ | $r_c, r_r, N_c, N_r$ |
| cold phase processes | | | |
| heterogeneous nucleation | HIND | $r_v, N_{IFN,CIN}$ | $r_i, N_i$ |
| autoconversion of pristine ice | CNVS | $r_i, N_i$ | $r_s, N_s$ |
| sublimation of aggregates | CNVI | $r_s, N_s$ | $r_i, N_i$ |
| aggregation of pristine ice | AGG | $r_i, N_i$ | $r_s$ |
| self collection of snow/aggregates | SC | $N_s$ | $\oslash$ |
| dry growth of the graupels | DRYG | $r_i, r_s, N_i, N_s$ | $r_g$ |
| wet growth of the graupels | WETG | $r_i, r_s, N_i, N_s$ | $r_g$ |
| conversion melting | CMEL | $r_s, N_s$ | $r_g, N_g$ |
| wet growth of hail | WETH | $r_c, r_i, r_s, r_g, N_c, N_i, N_s, N_g$ | $r_h, N_h$ |
| conversion of hail into graupel | COHG | $r_h, N_h$ | $r_g, N_g$ |
| deposition on ice crystals (/sublimation) | DEPI (/ SUB) | $r_v (/r_i, N_i)$ | $r_i (/r_v)$ |
| deposition on snow/aggregates (/sublimation) | DEPS (/ SUB) | $r_v (/r_s, N_s)$ | $r_s (/r_v)$ |
| deposition on graupel (/sublimation) | DEPG (/ SUB) | $r_v (/r_g, N_g)$ | $r_g (/r_v)$ |
| deposition on hail (/sublimation) | DEPH (/ SUB) | $r_v (/r_h, N_h)$ | $r_h (/r_v)$ |
| sedimentation | SEDI | $r_i, r_s, r_g, r_h, N_i, N_s, N_g, N_h$ | $r_i, r_s, r_g, r_h, N_i, N_s, N_g, N_h$ |
| mixed phase processes | | | |
| homogeneous nucleation | HONR | $r_r, N_r$ | $r_g, N_g$ |
| immersion freezing of coated IFN | IFR | $r_c, N_c$ | $r_i, N_i$ |
| raindrops contact freezing | CFRZ | $r_r, r_i, N_r, N_i$ | $r_g, N_g$ |
| light riming of aggregates | LRIM | $r_c, N_c$ | $r_s$ |
| heavy riming of aggregates | HRIM | $r_c, r_s, N_c, N_s$ | $r_g, N_g$ |
| accretion of rain and aggregates | ACC | $r_r, r_s, N_r, N_s$ | $r_s, r_g, N_g$ |
| Hallet-Mossop process | HMS-G | $r_s, r_g$ | $r_i, N_i$ |
| Homogeneous nucleation | HONC | $r_c (/ N_c)$ | $r_i, N_i$ |
| dry growth of the graupels | DRYG | $r_c, r_r, N_r, N_c$ | $r_g$ |
| partial freezing & water shedding | WETG & SHED | $r_c, r_r, N_r, N_c$ | $r_g, N_r$ |
| ice melting | IMLT | $r_i, N_i$ | $r_c, N_c$ |
| graupel melting | GMLT | $r_g, N_g$ | $r_r, N_r$ |
| hail melting | HMLT | $r_h, N_h$ | $r_r, N_r$ |

**Table A1.** List of symbols

| Symbols | Description | SI units |
|---|---|---|
| $a$ | Pre-factor of the mass-diameter relationship | $\mathrm{kg\,m^{-b}}$ |
| $a_g$ | Pre-factor of the graupel mass-diameter relationship | $\mathrm{kg\,m^{-b}}$ |
| $a_s$ | Pre-factor of the snow/aggregates mass-diameter relationship | $\mathrm{kg\,m^{-b}}$ |
| $a_y$ | Pre-factor of the mass-diameter relationship of species $y$ | $\mathrm{kg\,m^{-b}}$ |
| $b$ | Exponent of the mass-diameter relationship | |
| $b_g$ | Exponent of the graupel mass-diameter relationship | |
| $b_s$ | Exponent of the snow/aggregates mass-diameter relationship | |
| $b_x$ | Exponent of the mass-diameter relationship of species $x$ | |
| $b_y$ | Exponent the mass-diameter relationship of species $y$ | |
| $c$ | Pre-factor of the fall speed-diameter relationship | $\mathrm{m^{1-d}\,s^{-1}}$ |
| $c_y$ | Pre-factor of the fall speed-diameter relationship of species $y$ | $\mathrm{m^{1-d}\,s^{-1}}$ |
| $C$ | Pre-factor of the $N = C\lambda^x$ relationship | |
| $\mathcal{C}_1$ | Parameter of the capacitance-diameter relationship | |
| $d$ | Exponent of the fall-speed-diameter relationship | |
| $d_y$ | Exponent of the fall-speed-diameter relationship of species $y$ | |
| $D$ | Hydrometeor diameter | m |
| $D_r$ | Diameter of raindrop | m |
| $D_r^{lim}$ | Diameter beyond which raindrop-collecting aggregates are converted into graupel | m |
| $D_s$ | Diameter of snow/aggregate | m |
| $D_s^{lim}$ | Diameter beyond which riming aggregates are converted into graupel | m |
| $D_t$ | Diameter beyond which ice crystals are converted into snow aggregates | m |
| $D_x$ | Diameter of species $x$ | m |
| $D_y$ | Diameter of species $y$ | m |
| $D_y^{lim}$ | Diameter of species $y$ beyond which species $y$ is converted into species $z$ | m |
| $E_{cg}$ | Efficiency of graupel collecting cloud droplets | |
| $E_{cs}$ | Efficiency of snow collecting cloud droplets | |
| $E_{ir}$ | Efficiency of graupel collecting ice crystals | |
| $E_{ir}$ | Efficiency of raindrops collecting ice crystals | |
| $E_{is}$ | Efficiency of snow collecting ice crystals | |
| $E_{rg}$ | Efficiency of graupel collecting raindrops | |
| $E_{sg}$ | Efficiency of graupel collecting snow | |
| $E_{ss}$ | Efficiency of snow-snow collection | |
| $E_{xy}$ | Efficiency of species $x$ collecting species $y$ | |
| $g(D)$ | Generalized Gamma distribution law | |
| $g_x(D_x)$ | Generalized Gamma distribution law for species $x$ | |
| $g_y(D_y)$ | Generalized Gamma distribution law for species $y$ | |

**Table A2.** List of symbols (Continued)

| Symbols | Description | SI units |
|---|---|---|
| $K$ | Collection kernel | |
| $m(D)$ | Mass of hydrometeor of diameter $D$ | kg |
| $m_{0r}$ | Mass of a $0.72$ mm raindrop diameter | kg |
| $M(p)$ | p-order moment of the particle size distribution | |
| $M_x$ | Moment of the particle size distribution of species $x$ | |
| $M_y$ | Moment of the particle size distribution of species $y$ | |
| $n(D)$ | Number concentration of hydrometeors with $D <$ diameter $< D + dD$ | $\text{kg}^{-1}\,\text{m}^{-1}$ |
| $N$ | Number concentration of hydrometeors | $\text{kg}^{-1}$ |
| $N_r$ | Number concentration of raindrops | $\text{kg}^{-1}$ |
| $N_s$ | Number concentration of snow/aggregates | $\text{kg}^{-1}$ |
| $N_x$ | Number concentration of species $x$ | $\text{kg}^{-1}$ |
| $N_y$ | Number concentration of species $y$ | $\text{kg}^{-1}$ |
| $r$ | Mixing ratio | $\text{kg}\,\text{kg}^{-1}$ |
| $r_g$ | Mixing ratio of graupel | $\text{kg}\,\text{kg}^{-1}$ |
| $r_r$ | Mixing ratio of rain | $\text{kg}\,\text{kg}^{-1}$ |
| $r_s$ | Mixing ratio of snow/aggregates | $\text{kg}\,\text{kg}^{-1}$ |
| $r_x$ | Mixing ratio of species $x$ | $\text{kg}\,\text{kg}^{-1}$ |
| $r_y$ | Mixing ratio of species $y$ | $\text{kg}\,\text{kg}^{-1}$ |
| $T$ | Temperature | K |
| $T_t$ | Triple point temperature | K |
| $v$ | Hydrometeor terminal fall speed | $\text{m}\,\text{s}^{-1}$ |
| $v_x$ | Terminal fall speed of species $x$ | $\text{m}\,\text{s}^{-1}$ |
| $v_y$ | Terminal fall speed of species $y$ | $\text{m}\,\text{s}^{-1}$ |
| $x$ | Exponent of the $N = C\lambda^x$ relationship | |
| $\alpha$ | Parameter for the hydrometeors size distributions | |
| $\alpha_{s \to g}$ | Coefficient to transfer melting aggregates to graupel | |
| $\Gamma$ | Gamma function | |
| $\lambda$ | Slope parameter of the hydrometeors size distribution | $\text{m}^{-1}$ |
| $\lambda_x$ | Slope parameter of the size distribution of species $x$ | $\text{m}^{-1}$ |
| $\lambda_y$ | Slope parameter of the size distribution of species $y$ | $\text{m}^{-1}$ |
| $\lambda_x^{min}$ | Minimum value of the slope parameter of the size distribution of species $x$ used to compute the lookup tables | $\text{m}^{-1}$ |
| $\lambda_x^{max}$ | Maximum value of the slope parameter of the size distribution of species $x$ used to compute the lookup tables | $\text{m}^{-1}$ |
| $\Lambda_r(\lambda_x, \lambda_y)$ | Normalization factor of the mass collection kernel | |
| $\Lambda_N(\lambda_x, \lambda_y)$ | Normalization factor of the number concentration collection kernel | |
| $\nu$ | Parameter for the hydrometeors size distributions | |

**Table A3.** List of symbols (Continued)

| Symbols | Description | SI units |
|---|---|---|
| $\rho_a$ | Air density | $\mathrm{kg\,m^{-3}}$ |
| $\rho_{00}$ | Air density at the reference pressure level | $\mathrm{kg\,m^{-3}}$ |
| $\rho_s$ | Density of snow/aggregates | $\mathrm{kg\,m^{-3}}$ |
| $\rho_{sr}$ | Density of aggregate-raindrop mixture | $\mathrm{kg\,m^{-3}}$ |
| $\rho_w$ | Liquid water density | $\mathrm{kg\,m^{-3}}$ |
| $\Delta_{\mathrm{CMEL}} N_s$ | Number concentration tendency of snow due to conversion melting | $\mathrm{kg^{-1}\,s^{-1}}$ |
| $\Delta_{\mathrm{CNVS}} N_i$ | Number concentration tendency of ice crystals due to conversion to snow/aggregates | $\mathrm{kg^{-1}\,s^{-1}}$ |
| $\Delta_{\mathrm{CNVS}} N_s$ | Number concentration tendency of snow due to ice conversion to snow/aggregates | $\mathrm{kg^{-1}\,s^{-1}}$ |
| $\Delta_{\mathrm{CNVS}} r_i$ | Mixing ratio tendency of ice crystals due to conversion to snow/aggregates | $\mathrm{kg\,kg^{-1}\,s^{-1}}$ |
| $\Delta_{\mathrm{COL}} r_x$ | Mixing ratio tendency of species $x$ due to the mass collection of $y$ | $\mathrm{kg\,kg^{-1}\,s^{-1}}$ |
| $\Delta_{\mathrm{COL}} r_y$ | Mixing ratio tendency of species $y$ due to the mass collection of $x$ | $\mathrm{kg\,kg^{-1}\,s^{-1}}$ |
| $\Delta_{\mathrm{COL}} r_{x \to y}$ | Mixing ratio tendency of species $y$ from species $x$ when species $z$ can be formed | $\mathrm{kg\,kg^{-1}\,s^{-1}}$ |
| $\Delta_{\mathrm{COL}} r_{y \to z}$ | Mixing ratio tendency of species $z$ from species $x$ and $y$ | $\mathrm{kg\,kg^{-1}\,s^{-1}}$ |
| $\Delta_{\mathrm{COL}} N_x$ | Number concentration tendency of species $x$ due to the collection of $y$ | $\mathrm{kg^{-1}\,s^{-1}}$ |
| $\Delta_{\mathrm{COL}} N_y$ | Number concentration tendency of species $y$ due to the collection of $x$ | $\mathrm{kg^{-1}\,s^{-1}}$ |
| $\Delta_{\mathrm{COL}} N_{x \to y}$ | Number concentration tendency of species $y$ from species $x$ when species $z$ can be formed | $\mathrm{kg^{-1}\,s^{-1}}$ |
| $\Delta_{\mathrm{COL}} N_{y \to z}$ | Number concentration tendency of species $z$ from species $x$ and $y$ | $\mathrm{kg^{-1}\,s^{-1}}$ |
| $\Delta_{\mathrm{DRYG}} r_c$ | Mixing ratio tendency of cloud droplets due to graupel dry growth | $\mathrm{kg\,kg^{-1}\,s^{-1}}$ |
| $\Delta_{\mathrm{DRYG}} r_r$ | Mixing ratio tendency of raindrops due to graupel dry growth | $\mathrm{kg\,kg^{-1}\,s^{-1}}$ |
| $\Delta_{\mathrm{DRYG}} r_i$ | Mixing ratio tendency of ice crystals due to graupel dry growth | $\mathrm{kg\,kg^{-1}\,s^{-1}}$ |
| $\Delta_{\mathrm{DRYG}} r_s$ | Mixing ratio tendency of snow/aggregates due to graupel dry growth | $\mathrm{kg\,kg^{-1}\,s^{-1}}$ |
| $\Delta_{\mathrm{DRYG}} r_g$ | Mixing ratio tendency of graupel due to dry growth | $\mathrm{kg\,kg^{-1}\,s^{-1}}$ |
| $\Delta_{\mathrm{GMLT}} N_g$ | Number concentration tendency of graupel due to melting | $\mathrm{kg^{-1}\,s^{-1}}$ |
| $\Delta_{\mathrm{GMLT}} N_r$ | Number concentration tendency of raindrops due to graupel melting | $\mathrm{kg^{-1}\,s^{-1}}$ |
| $\Delta_{\mathrm{GMLT}} r_g$ | Mixing ratio tendency of graupel due to melting | $\mathrm{kg\,kg^{-1}\,s^{-1}}$ |
| $\Delta_{\mathrm{SMLT}} N_s$ | Number concentration tendency of snow due to melting | $\mathrm{kg^{-1}\,s^{-1}}$ |
| $\Delta_{\mathrm{SMLT}} r_s$ | Mixing ratio tendency of snow due to melting | $\mathrm{kg\,kg^{-1}\,s^{-1}}$ |
| $\Delta_{\mathrm{WETG}} N_c$ | Number concentration tendency of cloud droplets due to graupel wet growth | $\mathrm{kg^{-1}\,s^{-1}}$ |
| $\Delta_{\mathrm{WETG}} N_i$ | Number concentration tendency of ice crystals due to graupel wet growth | $\mathrm{kg^{-1}\,s^{-1}}$ |
| $\Delta_{\mathrm{WETG}} N_r$ | Number concentration tendency of raindrops due to graupel wet growth | $\mathrm{kg^{-1}\,s^{-1}}$ |
| $\Delta_{\mathrm{WETG}} N_s$ | Number concentration tendency of snow/aggregates due to graupel wet growth | $\mathrm{kg^{-1}\,s^{-1}}$ |
| $\Delta_{\mathrm{WETG}} r_r$ | Mixing ratio tendency of raindrops during graupel wet growth | $\mathrm{kg\,kg^{-1}\,s^{-1}}$ |
| $\Delta_{\mathrm{WETG}} r_h$ | Mixing ratio tendency of hail due to graupel wet growth | $\mathrm{kg\,kg^{-1}\,s^{-1}}$ |
| $\Delta v_{xy}$ | Scaled terminal fall speed difference in mass collection kernel | $\mathrm{m\,s^{-1}}$ |
| $\Delta v_{N,xy}$ | Scaled terminal fall speed difference in number concentration collection kernel | $\mathrm{m\,s^{-1}}$ |