# Peer review of "LIMA (v2.0): A full two-moment cloud microphysical scheme for the mesoscale non-hydrostatic model Meso-NH v5-6"

_EGUsphere, 2024_

## Referee Comment (RC1)

**Review of *LIMA (v2.0)***

Tarfour et al. present an extension of the "LIMA" microphysics scheme which adds prognostic number concentration to describe hail and ice hydrometeors. The additional process rates and assumptions are thoroughly documented, and the scheme is demonstrated with comparison to LIMA v1.0 in a deep convective cloud simulated in MesoNH. The source code is open source and documented for reproducing results. I have a few concerns related to figures, discussion, and the model description in this manuscript that warrant minor revisions, and otherwise believe it to be a strong contribution to the model development literature.

**Comments:**
1. One aspect of discussion is noticeably lacking in the introduction/conclusions: a comparison of the 2-moment prognostic approach in LIMA v2.0 with other existing 2-moment (and alternative) approaches. In particular I am thinking of the popular MG2008 and the P3 schemes. A few multimoment schemes are already mentioned in the introduction and citations, but without specific analogies to the new method presented. It would be ideal to understand specifically which prognostic variables and process rates are determined in a similar fashion to existing 2-moment schemes, and which ones have taken a more novel approach.
2. Several variables in section 3 are lacking a definition or reference. While many of these are free parameters that are listed in Table 1, others are not defined anywhere. For instance, what are...
    a. L182: the lambda_x range? Lambda is mentioned again in L505 without any reference to what this parameter corresponds to, or why it is important.
    b. Eq 7: rho_a (I assume air density?)
    c. Eq 13-15: free parameters a_y, b_y, c_y, d_y, rho_00, rho_dref, etc.
    d. L231 and Eq 17: T (air temperature or hydrometeor temperature?)
    e. Eq 19: g(D) ?
3. L290, the assumption of that "mean particle mass does not change", does not justified and should be clarified. For instance, the XMLT process leads to denser aggregates, which would imply that mean particle mass is increasing.
4. I would like to see a discussion of the complexity of v2.0 versus the single-moment v1.0. How many additional free parameters, prognostic variables, and process rates are required compared with the one-moment version? How does the time to run your 3D DCC simulation change with the addition of this complexity?
5. Several of the figures are very information dense, and could be improved to focus on specific quantities that are important to the new microphysics method. For instance:
    a. Figure 2 and 4: I suggest showing the hail precipitation/accumulation with colors (and either removing rain production, or using contour lines) rather than the patterns. Because LIMA v1 and v2 use the same prognostic variables for liquid microphysics, the differences in ice/hail hydrometeors is

the more interesting quantity, and is very difficult to see in the existing presentation.

b. I'm unclear why there is so much focus on updraft and downdraft magnitudes/locations in the results (ex. L351-355, Figure 3). My takeaway from figure 3 is that the simulations are dynamically similar (which one would expect since they use the same turbulence scheme and initialization), and thus differences in precipitation rates stem from microphysics, similar to the statement in L364-366. For this reason I believe you could eliminate the red/blue contours from figure 4 in order to make the results easier to read.

c. It is almost impossible to read the pie charts in figure 5, though I appreciate what the authors are trying to portray here. I think it would be more effective to display the full horizonal averages (hydrometeor concentration as a function of altitude) at the sacrifice of the isotherms and contours, which are challenging to read anyways. Then a more direct comparison between the altitude maxima and type of hydrometeors can be made for LIMA v1 and v2.

d. The legends/text in figure 8 should be made larger for readability.

6. In a few places, you mention that your results are "in line with conceptual schemes" (L348) or "in agreement with the observations" (L468), but it is not clear what these conceptual schemes state, or which observations are being compared. Please be more specific, especially since the conceptual scheme is mentioned again in L509.

7. In figure 9a indicates that the diagnostic relationship from LIMAv1 has an inverse relationship of snow mixing ratio and number concentration, contrary to the v2 results, which seems like a substantial difference in underlying assumptions. Can you address this discrepancy?

8. L468-475 provide an excellent summary of the key findings from the simulations!

9. The conclusions section could be improved to maintain focus on the findings of this study, and suggest a clear and specific path forward. In particular, the final two paragraphs focus on radar reflectivity and aerosol processing (which are not mentioned earlier in the paper) without making it clear which aspects of this future research are currently possible with LIMA v2.0, and which require further development. I suggest clearly stating that "future work is required" to enable comparison of radar reflectivity (and why it is a future metric), and that the prognostic number concentrations to LIMA 2.0 "enables future research" on aerosol impacts on hail and ice hydrometeors.

**Other language/typos:**

L11: "to produce" → "at producing"
L12: "to reduce" → "at reducing"
L151: what is a "releasable process" ? This wording doesn't make sense.
L182: "A new tables" → "New tables" or "A new table"
L302-305 is repeated twice (L206-L308)
L445: "dimensions of snow and graupel" ; what is the "dimension"? The mean size?
L447: "remain available" → "remains available"
L505: "observations Tarfour et al…" → "observations (Tarfour et al…)"

---

## Referee Comment (RC2)

Review of "LIMA (v2.0): A full two-moment cloud microphysical scheme for the mesoscale non-hydrostatic model Meso-NH v5-6" by Tanfour et al."

This study developed LIMA v2.0 worked by newly predicting the number concentration of precipitating ice hydrometeors of snow aggregates, graupel, and hail. In addition, the 2nd version of LIMA was tested in the case of mid-latitude supercell to examine the differences in the representation of ice particle growth in comparison to the 1st version of LIMA. The authors successfully presented the advantage of the full 2 moment framework with the focus on well-known issue on the number diagnosis in the 1 moment framework. The manuscript is well organized and the conclusion is well supported by the figures. However, readability is not a bit good. In addition, there exist some confusing descriptions. These points will be fixed soon with no additional analysis. Therefore, my recommendation is minor revision rather than major revision although the author needs to fix a lot of points.

Major points on readability

1.  In section 4.2, it is not clear whether the authors explain about LIMA or LIMA2. The authors should clarify "in LIMA2" or "in LIMA" after the explanation.

    1.1. Line#378, "whereas LIMA2" would be "whereas LIMA". Is it right?

    1.2. Line#397, this sentence is about "LIMA2"

    1.3. Line #418, "The onset is 6min later …" would be the explanation for LIMA2. Is its right? In addition, what do you mean with the words "6 min later"? What occurs before the onset?

    1.4. Line#427, the sentence begins with "However, the mixing ratios …" may be description about LIMA2. Is it right?

    1.5. Line#446-447, Which do you mention after the "leading to a reduction in .."? LIMA or LIMA2? Regarding the conjunction, it would be LIMA. However, in the context, it may be LIMA2. In addition, this sentence is confusing. Please rewrite the sentence.

    1.6. Line#447-449. This sentence is also confusing. Which process in LIMA2 do you indicate? I think aggregation of cloud ice by snow is "not" efficient in LIMA2 as was shown in Figure 8e. Therefore, large amount of cloud ice exists in LIMA2. What do you mean with the words of "the process is more efficient in the LIMA2"? Is it LIMA?

2.  Readability of figures is not good.

    2.1. Regarding Figure 2 and Figure 4, I don't understand what the stipple in the legend means. What is $x$ and what is the unit of $x$? Please describe them in the caption.

    2.2. Regarding the figure caption in Figure 4, "between 0 and 4 km" is to be clarified as "at altitudes from 0 to 4 km".

    2.3. Regarding Figure 5, it is difficult to distinguish the cloud ice color from the snow color in the

pie charts. Please use different color tone for snow to explain the differences in cloud ice and snow distribution in the main text.

2.4. Regarding Figure 8, I may miss the description of some processes. Where is HONC in (a)? In addition, it is difficult to distinguish the colors between SEDI and CNVI in (a) and between SEDI and HMLT in (d). In general, this panel is too busy to follow the important differences between LIMA and LIMA2. I suggest to pick up only the upper most 3 processes in each figure (a-h) and other minor processes to be omitted or shown in supporting information.

2.5. Regarding Figure 8 and table 1, CNVS is sublimation of aggregates. Thus, CNVS should be negative. However, CNVS is positive in (b). In addition, I don't know why sublimation of aggregates is shown in (a) although sublimation of snow does not affect cloud ice. I guess the description of CNVS is wrong. Please fixed table 1.

2.6. Regarding Figure 9, what is the unit of diameter isolines indicated by dashed grey lines. Is it mm?

2.7. In Table 2, HIND and SEDI are not included. CNVS may be wrong. In addition, I do not distinguish between CNVI and CNVS. What is the difference in the physical processes dividing CNVI and CNVS?

Specific comments.

1. In Abstract, the difference in the mechanisms between LIMA and LIMA2 were not described at all. Therefore, I suggest to emphasize the unreasonable diagnosis of number concentration of snow, graupel, and hail in the one moment framework and then briefly describe that the prediction of number concentration reasonably slow down the growing processes based on physics.

2. Line#386-8, How do you determine "advection is dominant"? I think this is the result and is not the cause. In LIMA2, the number concentration of snow and graupel significantly decrease by reasonable representation. Based on Section 3.1, the riming terms are proportional to the particle number concentration. This results in significant reduction in the removal of rain droplets by riming. In addition, prediction of rain droplets selectively removes larger rain droplets faster through riming and sedimentation. As a result, smaller rain droplets are likely to be transported upward due to smaller terminal velocity.

3. Equation (23), I require the documentation of the $\lambda_{min}$ and $\lambda_{max}$ for each hydrometeor in a table to hold the reproducibility of the model description. In addition, the number of cells used for the two-dimensional look up table is required too in the same table. One may follow your article to develop their own full 2-moment cloud microphysics scheme. The perfect way

is also documenting the accuracy of the look up table because the accuracy depends on the number of cells, but I don't require this level of documentation.

4. Line#415-432, The authors should mention the rationality of 2-moment schemes and evident errors in 1-moment schemes based on Figure 7.

For example, graupel is produced by freezing of rain droplets or riming of snow. Therefore, Ng should be equal to or smaller than Ns or Nr. However, Ng is significantly larger than Ns and Nr based on Figures 7b,c, and f. In addition, assuming a binary collision, Ns values is to be close to half of Ni when much snow is initiated by self-aggregation of cloud ice at t=20 min in Figures 6b and 7b. However, Ns is significantly smaller than Ni by four to five digits. Therefore, diagnoses of Ns and Ng are clearly wrong.

In addition, please show the references articles of the number diagnoses of Nr, Ns, Ng, and Nh. I guess that individual diagnosis was obtained in different types of rainfall systems. When Nr and Ng diagnoses are obtained in the same case, Ng would be smaller than Nr as was represented by LIMA2. However, if Nr diagnosis was obtained in maritime rain systems and Ng diagnosis was obtained in the continental supercells, Ng could be significantly larger than Nr. In this way, consistency among the diagnoses is important for one-moment schemes. Please discuss these points to emphasize the rationality of 2-moment schemes and deficiency in 1-moment schemes.

5. Line#462, I don't understand the context. Why is "nevertheless" used here? When different cases were observed, different diagnoses were obtained. In this manuscript, the objective rainfall system is provided by Klemp and Wilhelmson (1978) and is different from the system observed by Taufour et al. (2018). Therefore, it is obvious that the black line on Figure 9a does not exactly follows the major portion of the LIMA2 simulations.

Instead, the most important point of the figure is the similarity of the major relationships between mixing ration and number concentration. LIMA2 simulations show that number concentration increases as the mixing ratio increases. This feature is also observed in Taufour's observations. In LIMA2, the mean volume diameter gradually increases from 3.0 to 8.0 as the mixing ratio increases from $10^{-3}$ to $10^0$ g kg$^{-1}$, whereas the mean volume diameter increases from 0.9 to 2.0 in the same mixing ratio range. This indicates that Taufour's case is relatively moderate rainfall systems compared to Klemp and Wilhelmson's case based on the differences in the mean volume diameter.

6. The paragraph line#468 to #475 is to be put on conclusion section. Similarly, the paragraph from line#477 to #481 is described for the future work. Thus, that is not to be described in the

result section. I suggest to delete the paragraph. The sentences from line#501 to #506 should be modified as the future prospects based on this study. I guess the context as that full-2moment scheme can be utilized as a reference of the number diagnoses used in 1-moment schemes. The paragraph from line#507 to #512 should be moved to introduction section because this is not a summary nor a conclusion

Technical comments

1. Line#35, the sentence begins with "This type of scheme for…" is confusing. Please rewrite the sentence.

2. Line#55-57, the sentence begins with "Comparisons of these studies…" is confusing. please rewrite the sentence.

3. Line#59, what does "multi-moment diagram" mean?

4. Line#59, the word "shape parameters" is generally used for the parameter characterizes the shape of the particle size distribution as was used by the authors at line#121 with Eq. (1). I suggest to use "shape of nonspherical ice hydrometeors" or something.

5. Line#60, what does the sentence "the impact of …" means? Please rewrite the sentence.

6. Line#62, what do "the different schemes" indicate?

7. Line#65, what does the sentence "It is very likely …" means? Please rewrite the sentence.

8. Line#79, Please replace "Conversely" with "In contrast",

9. Eq.(4), the dimension of righthand side is wrong. I think $\overline{D}^p$ should be added because the r.h.s should be equal to $N\overline{D}$ with $p=1$.

10. Line#140, does "CND or EVP" mean the saturation adjustment based on this description? It is known to be better to solve condensation and evaporation explicitly as was solved for ice particles because the timescale of condensation/evaporation is sometimes larger than model timestep, particularly in regional simulations. You can easily find the discussion about aerosol condensation effect of something. This is just a comment.

11. Line#159-160, Doesn't homogeneous freezing of rain droplets turn into hail? I think hail is a dense frozen particle. Thus, frozen rain would be a kind of hail.

12. Section 3. Please show the difference in the calculation costs. I'd like to know an increase in the calculation cost of microphysics and increase in the total calculation cost.

13. Section 3.1.1, In addition to (I)-(III), (IV) self-aggregation of snow, graupel are necessary. In this case, mixing ratio does not change but number concentration reduces. This point is important particularly for the prediction of snow.

14. Eq. (13)-(15) in Section 3.1.2, why is $E_{xy}$ excluded from the integration? Does the term depend on size or mass as was proposed by Böhm (1999)? Please clarify the formulation of $E_{xy}$ here.

Böhm, J. P., 1999: Revision and clarification of "A general hydrodynamic theory for mixed-phase microphysics." Atmospheric Research, 52, 167–176, https://doi.org/10.1016/S0169-8095(99)00033-2.

15. Eq. (13)-(15) in Section 3.1.2, what is $\rho_{dref}$? Based on Eq. (3), that is to be $\rho_a$.

16. Eq. (16) contains undefined terms of $\Delta_{DRYG}r_c, \Delta_{DRYG}r_r, \Delta_{DRYG}r_i, and\ \Delta_{DRYG}r_g$. I suppose the terms as $\Delta_{COL}r_c(c-g), \Delta_{COL}r_r(r-g), \Delta_{COL}r_i(r-i), and\ \Delta_{COL}r_s(g-s)$ as was defined in the case (II) in Section 3.1.1. Is it right?

17. Title of Section 3.1.3., the word "significant" would be replaced with "non-negligible". Similarly, significant at line#247 would be done.

18. Line#247, please close the sentence before "and" to increase readability. Then, please start the sentence with "Therefore" instead of using "therefore" in the middle part of the sentence. In addition, it would better to insert "in this study" at the end of the sentence.

19. Line#265-273 (Rain accretion on aggregates ACC), do the diameters mean the diameter of individual particles or PSD mean diameter? If it means the diameter of individual particles, how do you integrate the collection kernel? Could you clarify this point?

20. Line#285, what is "a threshold"? Please clarify that.

21. Line#292, "primary ice crystal" would be a typo of "pristine ice crystal".

22. Line#292, $\Delta_{CNV}r_i$ has not been documented. Please document the equation of the growth term or please refer to the original article in which the term is documented.

23. Line#299, water formed "*on*" the surface

24. Line#302, here the authors assumed melting graupel particles are larger than 0.72 mm. Do you use something of a criterion for shedding graupel diameter?

25. Line#303-304, please refer to the articles, which document the wind-tunnel experiments.

26. Line#361, At first glance, I don't understand what do "30 (3.5 mm/h)" means. To increase readability, it is better to modify as "30 min (3.5 mm/h), 34 min (4.5mm/hr) …".

27. Line#398, this sentence is not necessary.

28.  Line#412, you should remove "it seems to be" because this point is evident from the figure. In addition, it is better to mention the difference in snow amount. Since snow is produced by aggregation of cloud ice, large amount of snow indicates the rapid consumption of cloud ice. This point is clearly shown by Figure 8e.

29. Line#420, after "30 minutes" would be "40 minutes".

30. Line#423, what hypothesis do you mention here. I guess that is an issue in the number diagnosis. Please clarify that.

31. Line#452, In general, "not shown" is used when it is not important and it does not change

conclusion. When the authors did not show the figure, the results were not verified. Thus, please do not use "verified" here. I suggest to remove the sentence or add the figure. I think that point was found in Figures 8a-d, so you can refer to the figure in this sentence.

32. Line#462, I don't understand the wording "for the benefit" here. Isn't it deficient? Wrong diagnosis of the number concentrations results in wrong estimation of radiative properties. I think this is the deficient in 1-moment schemes.

33. Line#502, the wording "decoupling" does not match the context because the number concentration and mixing ratio should be coupled through the physical processes as was represented by LIMA2. Please change the wording.

---

## Author Response (AR1)

**Reply on RC1**

*Tarfour et al. present an extension of the "LIMA" microphysics scheme which adds prognostic number concentration to describe hail and ice hydrometeors. The additional process rates and assumptions are thoroughly documented, and the scheme is demonstrated with comparison to LIMA v1.0 in a deep convective cloud simulated in MesoNH. The source code is open source and documented for reproducing results. I have a few concerns related to figures, discussion, and the model description in this manuscript that warrant minor revisions, and otherwise believe it to be a strong contribution to the model development literature.*

We thank the reviewer for his/her time and efforts in reviewing our manuscript. The responses to his/her comments are addressed below.

**Comments:**

1. *One aspect of discussion is noticeably lacking in the introduction/conclusions: a comparison of the 2-moment prognostic approach in LIMA v2.0 with other existing 2-moment (and alternative) approaches. In particular I am thinking of the popular MG2008 and the P3 schemes. A few multimoment schemes are already mentioned in the introduction and citations, but without specific analogies to the new method presented. It would be ideal to understand specifically which prognostic variables and process rates are determined in a similar fashion to existing 2-moment schemes, and which ones have taken a more novel approach.*

   A paragraph presenting the MG2008 and P3 has been added in the introduction of the revised version of the manuscript: "To be complete, it is worth mentioning the novel parameterizations of Morrison and Grabowski (2008) and the P3 (Predicted Particle Properties) scheme (Morrison and Milbrandt, 2015; Milbrandt et al., 2021) where emphasis is put on the prediction of particle properties compared to the more classical approach based on predefined water and ice categories and providing conversion rates. Here our choice is to keep the classical point of view (LIMA like) in order to test an improvement of the whole scheme by considering a set of prognostic equations describing the number concentrations of precipitating ice particles."

2. *Several variables in section 3 are lacking a definition or reference. While many of these are free parameters that are listed in Table 1, others are not defined anywhere. For instance, what are…*

   1. *L182: the lambda_x range? Lambda is mentioned again in L505 without any reference to what this parameter corresponds to, or why it is important.*

As defined in Equation 2 and Line 123, $\lambda$ is the slope parameter of the size distribution. Thus $\lambda$min and $\lambda$max are the minimum and maximum values of the slope parameter used to precompute the integrals involving the collection kernels.

The paragraph dealing about the [$\lambda$min, $\lambda$max] range (lines 175-183 in the submitted manuscript) has been moved to section 3.1.3 in the new version of the manuscript.

2. *Eq 7: rho_a (I assume air density?)*

Yes, $\rho$a is the air density. It is clarified in the new version of the manuscript, from the first occurrence of this parameter.

3. *Eq 13-15: free parameters a_y, b_y, c_y, d_y, rho_00, rho_dref, etc.*

$\rho$00 is the air density at the reference pressure as defined line 128. *ay, by, cy* and *dy* are the free parameters of the mass-diameter and velocity-diameter relationships for species y (as defined at lines 125 and in Table 1). $\rho$dref has been changed to $\rho$a in the revised manuscript.

4. *L231 and Eq 17: T (air temperature or hydrometeor temperature?)*

T is air temperature.

5. *Eq 19: g(D) ?*

You are right: *g(D)* was not defined. *g(D)* is the normalized generalized Gamma distribution, so that *n(D) = N g(D)*. It is clarified in the new version of the manuscript.

In addition, a list of symbols is added as an appendix in the new version of the manuscript.

3. *L290, the assumption of that "mean particle mass does not change", does not justified and should be clarified. For instance, the XMLT process leads to denser aggregates, which would imply that mean particle mass is increasing.*

We realized that this sentence was not clear. It has been removed in the revised version of the manuscript, and the parameterization of processes other than collection is described in each paragraph.

4. *I would like to see a discussion of the complexity of v2.0 versus the single-moment v1.0. How many additional free parameters, prognostic variables, and process rates are required compared with the one-moment version? How does the time to run your 3D DCC simulation change with the addition of this complexity?*

In lines 166-172 of the first version of the manuscript, there was a short discussion about the complexity of LIMA v2.0 vs LIMA v1.0. It is stated that "for processes related to snow, graupel and hail already handled in LIMA v1.0, a new prognostic equation is added to the existing routines for handling number concentration transfer rates. For processes newly

handled in version 2.0, typically the self-collection of snow, a new routine is created including the parameterization of this process and called up in the LIMA monitor routine." The microphysical transfer rates newly handled in the v2.0 are plotted in red in Figure 1. Moreover, Table 2 gives the list of the microphysical processes in LIMA v2.0. If $Ns$, $Ng$ or $Nh$ is identified as a sink or a source, it means that a transfer rate for $Ns$, $Ng$ or $Nh$ is specifically computed for this process (and it was not the case in LIMA v1.0). Therefore 18 additional transfer rates are considered in LIMA v2.0.

To run this 3D storm, there is a 17 % increase in cpu/elapsed time when moving from v1.0 to v2.0. A paragraph has been added at the end of Section 4 to give more information about the numerical cost of LIMA v2.0. It is not only the added complexity in the microphysics scheme that is responsible for this increase in the computation time. Additional processes in the microphysics scheme make the cpu time increase by 9.7% when LIMA v2.0 is used instead of LIMA v1.0. When the full 2-moment scheme is used, the additional numerical cost is mainly attributed to the increase in the number of prognostic variables that must be forced, transported (advection and turbulence), exchanged and stored…

5. *Several of the figures are very information dense, and could be improved to focus on specific quantities that are important to the new microphysics method. For instance:*

   1. *Figure 2 and 4: I suggest showing the hail precipitation/accumulation with colors (and either removing rain production, or using contour lines) rather than the patterns. Because LIMA v1 and v2 use the same prognostic variables for liquid microphysics, the differences in ice/hail hydrometeors is the more interesting quantity, and is very difficult to see in the existing presentation.*

      Figures 2 and 4 have been modified in the new version of the manuscript. We changed the color schemes in order to increase the readability of the figure, and to improve their accessibility for readers with color vision deficiencies. In Figure 2, the black contours that represent hail precipitation at the ground are now more visible. In Figure 4, the color scheme for the precipitation rate was modified, and the number of colors and contours significantly decreased. We also changed the colors of the vertical wind speed to make hail rate more visible.

   2. *I'm unclear why there is so much focus on updraft and downdraft magnitudes/locations in the results (ex. L351-355, Figure 3). My takeaway from figure 3 is that the simulations are dynamically similar (which one would expect since they use the same turbulence scheme and initialization), and thus differences in precipitation rates stem from microphysics, similar to the statement in L364-366. For this reason I believe you could eliminate the red/blue contours from figure 4 in order to make the results easier to read.*

As we said in our response to the previous comment, Figure 4 has been modified to make instantaneous hail and precipitation rates more visible. However, we think it is important to show how updrafts and downdrafts are spatially distributed in the domain even if the LIMA and LIMA2 simulations are dynamically similar.

3. *It is almost impossible to read the pie charts in figure 5, though I appreciate what the authors are trying to portray here. I think it would be more effective to display the full horizonal averages (hydrometeor concentration as a function of altitude) at the sacrifice of the isotherms and contours, which are challenging to read anyways. Then a more direct comparison between the altitude maxima and type of hydrometeors can be made for LIMA v1 and v2.*

As also asked by the second reviewer, this figure has been modified. We did not change the representation using pie charts, but we suppressed unused information (the -10, -20 and -30°C isotherms, and the wind vectors), and changed the color schemes to increase readability. We also decreased the pie charts density but increased their size. The objective of this figure is to show the difference in the distribution of hydrometeors in the two simulations.

4. *The legends/text in figure 8 should be made larger for readability.*

The legends and text in figure 8 have been made larger. The color scheme has also been changed to increase readability, and only the processes with significant transfer rate have been plotted.

In general, to improve the accessibility of color figures, the color schemes of Figures 2, 4, 5, 6, 7, 8 and 9 have been modified. Color schemes described in Crameri et al. (2020) have been used when continuous palettes are needed (Figures 2, 6, 7 and 9), while the Okabe-Ito palette has been used for Figures 5 and 8.

6. *In a few places, you mention that your results are "in line with conceptual schemes" (L348) or "in agreement with the observations" (L468), but it is not clear what these conceptual schemes state, or which observations are being compared. Please be more specific, especially since the conceptual scheme is mentioned again in L509.*

When writing about the "conceptual scheme", we wanted to say that LIMA v2.0 enables representing a smaller hail pattern at the ground than LIMA v1.0. Moreover, as shown in Figure 4d, the hail pattern is decoupled from the updraft. This behavior seems more realistic when compared to compilations of observations (Doswell and Burgess, 1993 ; Kumjian and Ryzhkov, 2008). This has been specified in the revised version of the manuscript.

The term "in agreement with observations" was unnecessary in this sentence, and has been removed in the new version of the manuscript.

7. *In figure 9a indicates that the diagnostic relationship from LIMAv1 has an inverse relationship of snow mixing ratio and number concentration, contrary to the v2 results, which seems like a substantial difference in underlying assumptions. Can you address this discrepancy?*

The LIMA2 simulation exhibits a large variability of the $N$-$r$ couple. However, the maximum frequency of the $(N,r)$ couples has the same slope as the black line from Taufour et al. (2018), but shifted by one order of magnitude toward lower snow number concentration. In the LIMA simulation, the $(N,r)$ couple follows the green line in Figure 9a. It is clearly shown that the values assumed for $(C,x)$ in LIMA v1.0 lead to an inverse relationship between snow number concentration and snow mixing ratio in this case study as in Taufour et al. (2018).

In LIMA v1.0, the number concentration of snow/aggregates, graupel and hail is estimated using the relationship $N = C\lambda x$, where $\lambda$ depends on the mixing ratio $r$. Compiling results from the literature, Caniaux (1993) showed that $C$ and $x$ are linked by the relationship: $logC = -3.55x + 7.4$. Then, through sensitivity studies and physical considerations, he determined that the best couple $(C,x)$ for snow/aggregates was (5,2). However, in the 1-moment (ICE3, Pinty and Jabouille, 1998) and partial 2-moment (LIMA v1.0, Vié et al., 2016) schemes of Meso-NH, $x$ was set to 1 because taking $x$ too close to 2 would lead to some inconsistencies in computing $\lambda$. A positive value of $x$ enables to grossly reproduce the broadening of the spectra (a decrease of $\lambda$) by the self-aggregation processes when only mixing ratio is prognostic (LIMA v1.0). This statement has been added in Section 2.2 of the revised manuscript.

8. *L468-475 provide an excellent summary of the key findings from the simulations!*

Thank you! As recommended by the second reviewer, this paragraph has been moved to the conclusion.

9. *The conclusions section could be improved to maintain focus on the findings of this study, and suggest a clear and specific path forward. In particular, the final two paragraphs focus on radar reflectivity and aerosol processing (which are not mentioned earlier in the paper) without making it clear which aspects of this future research are currently possible with LIMA v2.0, and which require further development. I suggest clearly stating that "future work is required" to enable comparison of radar reflectivity (and why it is a future metric), and that the prognostic number concentrations to LIMA 2.0 "enables future research" on aerosol impacts on hail and ice hydrometeors.*

The conclusion has been modified in order to better identify the main results of this study and its perspectives.

**Other language/typos:**

*L11: "to produce" → "at producing"*

done

*L12: "to reduce" → "at reducing"*

done

*L151: what is a "releasable process" ? This wording doesn't make sense.*

These are processes that you can choose whether or not to activate. This is made clear in the new version of the manuscript.

*L182: "A new tables" → "New tables" or "A new table"*

This is corrected in the new version of the manuscript.

*L302-305 is repeated twice (L206-L308)*

You are right: this is corrected in the new version of the manuscript.

*L445: "dimensions of snow and graupel" ; what is the "dimension"? The mean size?*

Yes, in this sentence the dimension of the particles corresponds to the mean size. It is made clear in the new version of the manuscript.

*L447: "remain available" → "remains available"*

corrected

*L505: "observations Tarfour et al..." → "observations (Tarfour et al...)"*

done

**References**

Caniaux, G.: Paramétrisation de la phase glace dans un modèle non hydrostatique de nuage : application à une ligne de grains tropicale. 257pp., PhD thesis, Paul Sabatier University, Toulouse 3, 1993.

Crameri, F.: Scientific color maps, http://doi.org/10.5281/zenodo.1243862, 2018.

Crameri, F., Shephard, G., and Heron, P.: The misuse of color in science communication, Nat. Commun., 11, https://doi.org/10.1038/s41467-020-19160-7, 2020.

Doswell III, C. A. and Burgess, D. W.: Tornadoes and Tornadic Storms: a Review of Conceptual Models, p. 161–172, American Geophysical Union (AGU), https://doi.org/10.1029/GM079p0161, 1993.

Ducrocq, V., Braud, I., Davolio, S., Ferretti, R., Flamant, C., Jansa, A., Kalthoff, N., Richard, E., Taupier-Letage, I., Ayral, P.A., Belamari, S., Berne, A., Borga, M., Boudevillain, B., Bock, O., Boichard, J.L., Bouin, M.N., Bousquet, O., Bouvier, C., Chiggiato, J., Cimini, D., Corsmeier, U., Coppola, L., Cocquerez, P., Defer, E., Delanoë, J., Girolamo, P.D., Doerenbecher, A., Drobinski, P.,

Dufournet, Y., Fourrié, N., Gourley, J.J., Labatut, L., Lambert, D., Coz, J.L., Marzano, F.S., Molinié, G., Montani, A., Nord, G., Nuret, M., Ramage, K., Rison, W., Roussot, O., Said, F., Schwarzenboeck, A., Testor, P., Baelen, J.V., Vincendon, B., Aran, M. and Tamayo, J.: HyMeX-SOP1: the field campaign dedicated to heavy precipitation and flash flooding in the northwestern Mediterranean. Bulletin of the American Meteorological Society, 95(7), 1083–1100. https://doi.org/10.1175/BAMS-D-12-00244.1, 2014.

Kumjian, M. R. and Ryzhkov, A. V.: Polarimetric Signatures in Supercell Thunderstorms, J. Appl. Meteorol. Clim., 47, 1940–1961, https://doi.org/10.1175/2007JAMC1874.1, 2008.

Milbrandt, J. A., H. Morrison, D. T. Dawson II, and M. Paukert: A Triple-Moment Representation of Ice in the Predicted Particle Properties (P3) Microphysics Scheme. J. Atmos. Sci., 78, 439–458, https://doi.org/10.1175/JAS-D-20-0084.1, 2021.

Morrison, H., and W. W. Grabowski: A Novel Approach for Representing Ice Microphysics in Models: Description and Tests Using a Kinematic Framework. J. Atmos. Sci., 65, 1528–1548, https://doi.org/10.1175/2007JAS2491.1, 2008.

Morrison, H., and J. A. Milbrandt: Parameterization of Cloud Microphysics Based on the Prediction of Bulk Ice Particle Properties. Part I: Scheme Description and Idealized Tests. J. Atmos. Sci., 72, 287–311, https://doi.org/10.1175/JAS-D-14-0065.1, 2015.

Pinty, J. and Jabouille, P.: A mixed-phase cloud parameterization for use in mesoscale non hydrostatic model: simulations of a squall line and of orographic precipitations. In: Proceedings of Conference on Cloud Physics, Everett, WA, pp. 217–220. American Meteorological Society, Boston, MA, 2018.

Vié, B., Pinty, J.-P., Berthet, S., and Leriche, M.: LIMA (v1.0): A quasi two-moment microphysical scheme driven by a multimodal population of cloud condensation and ice freezing nuclei, Geosci. Model Dev., 9, 567–586, https://doi.org/10.5194/gmd-9-567-2016, 2016.

**Reply on RC2**
* * *
*Review of "LIMA (v2.0): A full two-moment cloud microphysical scheme for the mesoscale non-hydrostatic model Meso-NH v5-6" by Tanfour et al."*

*This study developed LIMA v2.0 worked by newly predicting the number concentration of precipitating ice hydrometeors of snow aggregates, graupel, and hail. In addition, the 2nd version of LIMA was tested in the case of mid-latitude supercell to examine the differences in the representation of ice particle growth in comparison to the 1st version of LIMA. The authors successfully presented the advantage of the full 2 moment framework with the focus on well-known issue on the number diagnosis in the 1 moment framework. The manuscript is well organized and the conclusion is well supported by the figures. However, readability is not a bit good. In addition, there exist some confusing descriptions. These points will be fixed soon with no additional analysis. Therefore, my recommendation is minor revision rather than major revision although the author needs to fix a lot of points.*

We thank the reviewer for her time and efforts in reviewing our manuscript. The responses to her comments are addressed below.

**Major points on readability**

1. *In section 4.2, it is not clear whether the authors explain about LIMA or LIMA2. The authors should clarify "in LIMA2" or "in LIMA" after the explanation.*

    1. *Line#378, "whereas LIMA2" would be "whereas LIMA". Is it right?*

        Yes, this is right. It is corrected in the new version of the manuscript.

    2. *Line#397, this sentence is about "LIMA2"*

        Yes, it is clarified in the new version of the manuscript.

    3. *Line #418, "The onset is 6min later ..." would be the explanation for LIMA2. Is its right? In addition, what do you mean with the words "6 min later"? What occurs before the onset?*

        This statement has been removed in the new version of the manuscript. Snow/aggregates mass starts to be produced at the same time in the LIMA and LIMA2 simulations as shown by the black solid line representing the maximum mixing ratio of snow in Figures 6b and 6d. However, a larger amount of snow mass is present in LIMA compared to LIMA2 between 15 and 30 min. As shown later in the microphysical budgets (Figure 8), LIMA tends to rapidly convert ice crystals into snow/aggregates.

4. *Line#427, the sentence begins with "However, the mixing ratios ..." may be description about LIMA2. Is it right?*

Yes, this is clarified in the new version of the manuscript.

5. *Line#446-447, Which do you mention after the "leading to a reduction in .."? LIMA or LIMA2? Regarding the conjunction, it would be LIMA. However, in the context, it may be LIMA2. In addition, this sentence is confusing. Please rewrite the sentence.*

6. *Line#447-449. This sentence is also confusing. Which process in LIMA2 do you indicate? I think aggregation of cloud ice by snow is "not" efficient in LIMA2 as was shown in Figure 8e. Therefore, large amount of cloud ice exists in LIMA2. What do you mean with the words of "the process is more efficient in the LIMA2"? Is it LIMA?*

We agree the sentences between line 446 and line 449 (comments 1.5 and 1.6) were confusing. This paragraph has been rewritten in the revised version of the manuscript.

2. *Readability of figures is not good.*

1. *Regarding Figure 2 and Figure 4, I don't understand what the stipple in the legend means. What is x and what is the unit of x? Please describe them in the caption.*

In the legend of Figures 2 and 4, $x$ corresponds to the hail rate at the ground. However, these two figures have been modified in the new version of the manuscript. We changed the color schemes in order to increase the readability of the figures, and to improve their accessibility for readers with color vision deficiencies. In Figure 2, the black contours that represent hail precipitation at the ground are now more visible. In Figure 4, the color scheme for the precipitation rate was modified, and the number of colors and contours significantly decreased. We also changed the colors of the vertical wind speed to make hail rate more visible.

2. *Regarding the figure caption in Figure 4, "between 0 and 4 km" is to be clarified as "at altitudes from 0 to 4 km".*

Done.

3. *Regarding Figure 5, it is difficult to distinguish the cloud ice color from the snow color in the pie charts. Please use different color tone for snow to explain the differences in cloud ice and snow distribution in the main text.*

This figure has been modified. We did not change the representation using pie charts, but we suppressed unused information (the -10, -20 and -30°C isotherms, and the wind vectors), and changed the color schemes to increase readability. We also decreased the pie charts density but increased their size.

4. *Regarding Figure 8, I may miss the description of some processes. Where is HONC in (a)? In addition, it is difficult to distinguish the colors between SEDI and CNVI in (a)*

*and between SEDI and HMLT in (d). In general, this panel is too busy to follow the important differences between LIMA and LIMA2. I suggest to pick up only the upper most 3 processes in each figure (a-h) and other minor processes to be omitted or shown in supporting information.*

In the revised version of the manuscript, microphysical processes that are not significant in the budget are not drawn. We also changed the color scheme to increase readability, and increased the text in the legend.

5. *Regarding Figure 8 and table 1, CNVS is sublimation of aggregates. Thus, CNVS should be negative. However, CNVS is positive in (b). In addition, I don't know why sublimation of aggregates is shown in (a) although sublimation of snow does not affect cloud ice. I guess the description of CNVS is wrong. Please fixed table 1.*

There was an error in Table 2 (see our response to comment 2.7) which has been corrected in the revised version of the manuscript.

6. *Regarding Figure 9, what is the unit of diameter isolines indicated by dashed grey lines. Is it mm?*

Yes, the unit of the mean volume diameter is mm. It has been added in the legend of Figure 9.

7. *In Table 2, HIND and SEDI are not included. CNVS may be wrong. In addition, I do not distinguish between CNVI and CNVS. What is the difference in the physical processes dividing CNVI and CNVS?*

To be consistent with Figure 8, HEN has been changed to HIND in Table 2. You are right, CNVS corresponds to the conversion of ice crystals into snow/aggregates while CNVI corresponds to the conversion of snow/aggregates into ice crystals: this has been corrected in the revised version of the manuscript. The sedimentation process (SEDI) has also been added in Table 2.

**Specific comments.**

1. *In Abstract, the difference in the mechanisms between LIMA and LIMA2 were not described at all. Therefore, I suggest to emphasize the unreasonable diagnosis of number concentration of snow, graupel, and hail in the one moment framework and then briefly describe that the prediction of number concentration reasonably slow down the growing processes based on physics.*

Several sentences have been added in the Abstract to emphasize the unreasonable diagnosis of number concentration of precipitating ice particles in LIMA v1.0, and to emphasize the slowing down of the growing processes of snow and graupel.

2. *Line#386-8, How do you determine "advection is dominant"? I think this is the result and is not the cause. In LIMA2, the number concentration of snow and graupel significantly decrease by reasonable representation. Based on Section 3.1, the riming terms are proportional to the particle number concentration. This results in significant reduction in the removal of rain droplets by riming. In addition, prediction of rain droplets selectively removes larger rain droplets faster through riming and sedimentation. As a result, smaller rain droplets are likely to be transported upward due to smaller terminal velocity.*

You are right. These two sentences have been removed from this paragraph since physical processes responsible for the differences in hydrometeors distribution are discussed a few paragraphs later.

3. *Equation (23), I require the documentation of the λmin and λ max for each hydrometeor in a table to hold the reproducibility of the model description. In addition, the number of cells used for the two-dimensional look up table is required too in the same table. One may follow your article to develop their own full 2-moment cloud microphysics scheme. The perfect way is also documenting the accuracy of the look up table because the accuracy depends on the number of cells, but I don't require this level of documentation.*

A few sentences have been added in Section 3.1.3 to document λmin and λmax for each hydrometeor. It is also made clear that an order of magnitude of λ is discretized over 10 points.

4. *Line#415-432, The authors should mention the rationality of 2-moment schemes and evident errors in 1-moment schemes based on Figure 7. For example, graupel is produced by freezing of rain droplets or riming of snow. Therefore, Ng should be equal to or smaller than Ns or Nr. However, Ng is significantly larger than Ns and Nr based on Figures 7b,c, and f. In addition, assuming a binary collision, Ns values is to be close to half of Ni when much snow is initiated by self-aggregation of cloud ice at t=20 min in Figures 6b and 7b. However, Ns is significantly smaller than Ni by four to five digits. Therefore, diagnoses of Ns and Ng are clearly wrong. In addition, please show the references articles of the number diagnoses of Nr, Ns, Ng, and Nh. I guess that individual diagnosis was obtained in different types of rainfall systems. When Nr and Ng diagnoses are obtained in the same case, Ng would be smaller than Nr as was represented by LIMA2. However, if Nr diagnosis was obtained in maritime rain systems and Ng diagnosis was obtained in the continental supercells, Ng could be significantly larger than Nr. In this way, consistency among the diagnoses is important for one-moment schemes. Please discuss these points to emphasize the rationality of 2-moment schemes and deficiency in 1-moment schemes.*

A paragraph has been added in Section 2.2 to discuss the diagnosis of $N_s$, $N_g$ and $N_h$. "In LIMA v1.0, the number concentration of snow/aggregates, graupel and hail is estimated using the relationship $N = C\lambda x$, where $\lambda$ depends on the mixing ratio $r$. This assumption for

aggregates implicitly takes into account the broadening of particle spectra to represent coalescence, and also implicitly treats the aggregation process. But these conditions are verified if: (i) $0 < x$, i.e. roughly reproduce the broadening of spectra (when $\lambda$ decreases) by self-aggregation processes ($N$ decreases), (ii) $x < b$, i.e. if the spectrum broadens as the snow/aggregates mixing ratio increases. Compiling results from the literature for snow, and graupel, Caniaux (1993) showed that $C$ and $x$ are linked by the relationship: $logC = -3.55x + 7.4$. Then, through sensitivity studies and physical considerations, he determined that the best couple ($C,x$) for snow/aggregates was (5,2). However, in the 1-moment (ICE3, Pinty and Jabouille, 1998) and partial 2-moment (LIMA v1.0, Vié et al., 2016) schemes of Meso-NH, $x$ was set to 1 because taking $x$ too close to 2 would lead to some inconsistencies in computing $\lambda$."

In the paragraphs dedicated to the presentation and discussion of Figure 7, a few sentences have been added to discuss the differences between the diagnosed and prognosed number concentration for snow, graupel and hail:

- "Since snow/aggregates are initiated by autoconversion of pristine ice and grow by aggregation of pristine ice or self-collection, it is expected that the number concentration of snow is less than (but close to) the number concentration of ice crystals. Indeed, in LIMA2, $Ni$ is about one order of magnitude higher than $Ns$. However, in LIMA, $Ni$ is five orders of magnitude higher than $Ns$ suggesting that the diagnostics of $Ns$ is not appropriate. "

- "As shown in Fig. 1 and Table 2, graupel is mainly produced by freezing of rain drops, riming of snow or accretion of rain and aggregates. Therefore, $Ng$ should be equal to or smaller than $Ns$ or $Nr$. This is the case in the LIMA2 simulation. However, Fig. 7 clearly shows that $Ng$ is significantly larger than $Ns$ and $Nr$ in the LIMA simulation."

5. *Line#462, I don't understand the context. Why is "nevertheless" used here? When different cases were observed, different diagnoses were obtained. In this manuscript, the objective rainfall system is provided by Klemp and Wilhelmson (1978) and is different from the system observed by Taufour et al. (2018). Therefore, it is obvious that the black line on Figure 9a does not exactly follows the major portion of the LIMA2 simulations. Instead, the most important point of the figure is the similarity of the major relationships between mixing ration and number concentration. LIMA2 simulations show that number concentration increases as the mixing ratio increases. This feature is also observed in Taufour's observations. In LIMA2, the mean volume diameter gradually increases from 3.0 to 8.0 as the mixing ratio increases from 10 -3 to 100 g kg-1, whereas the mean volume diameter increases from 0.9 to 2.0 in the same mixing ratio range. This indicates that Taufour's case is relatively moderate*

> *rainfall systems compared to Klemp and Wilhelmson's case based on the differences in the mean volume diameter.*

These three sentences have been removed in the revised version of the manuscript.

6. *The paragraph line#468 to #475 is to be put in the conclusion section. Similarly, the paragraph from line#477 to #481 is described for future work. Thus, that is not to be described in the result section. I suggest deleting the paragraph. The sentences from line#501 to #506 should be modified as the future prospects based on this study. I guess the context is that the full-2moment scheme can be utilized as a reference of the number diagnoses used in 1-moment schemes. The paragraph from line#507 to #512 should be moved to the introduction section because this is not a summary nor a conclusion.*

The text has been reorganized as suggested by the reviewer: the paragraph between line 468 and line 475 has been moved in the conclusion, while the paragraphs between line 507 and line 512 and between line 477 and line 481 have been removed in the new version of the manuscript.

**Technical comments**

1. *Line#35, the sentence begins with "This type of scheme for..." is confusing. Please rewrite the sentence.*

Done.

2. *Line#55-57, the sentence begins with "Comparisons of these studies..." is confusing. please rewrite the sentence.*

We wanted to say that multi-moment schemes provide greater variability and precision in hydrometeor size and reflectivity. This sentence has been rewritten.

3. *Line#59, what does "multi-moment diagram" mean?*

This is an error: it has been changed to "multi-moment schemes".

4. *Line#59, the word "shape parameters" is generally used for the parameter characterizes the shape of the particle size distribution as was used by the authors at line#121 with Eq. (1). I suggest to use "shape of nonspherical ice hydrometeors" or something.*

Done.

5. *Line#60, what does the sentence "the impact of ..." means? Please rewrite the sentence.*

You are right, this sentence is not clear. It has been clarified in the new version of the manuscript.

6. *Line#62, what do "the different schemes" indicate?*

"Different schemes" indicates "different microphysical schemes". It has been changed in the revised manuscript.

7. *Line#65, what does the sentence "It is very likely ..." means? Please rewrite the sentence.*

This sentence was not clear. It has been removed.

8. *Line#79, Please replace "Conversely" with "In contrast"*

Done

9. *Eq.(4), the dimension of righthand side is wrong. I think Dp should be added because the r.h.s should be equal to ND with p=1.*

There was an error in Eq. (4). It has been modified in the revised manuscript.

10. *Line#140, does "CND or EVP" mean the saturation adjustment based on this description? It is known to be better to solve condensation and evaporation explicitly as was solved for ice particles because the timescale of condensation/evaporation is sometimes larger than model timestep, particularly in regional simulations. You can easily find the discussion about aerosol condensation effect of something. This is just a comment.*

You are right. "CND/EVP" refers to the saturation adjustment as described in the LIMA v1.0 reference paper (Vié et al., 2016).

11. *Line#159-160, Doesn't homogeneous freezing of rain droplets turn into hail? I think hail is a dense frozen particle. Thus, frozen rain would be a kind of hail.*

In LIMA, homogeneous freezing of raindrops turns into graupel only. Hail is only formed from graupel wet growth in our scheme. This could be improved in a future version of LIMA.

12. *Section 3. Please show the difference in the calculation costs. I'd like to know an increase in the calculation cost of microphysics and increase in the total calculation cost.*

To run this 3D storm, there is a 17 % increase in cpu/elapsed time when moving from v1.0 to v2.0. A paragraph has been added at the end of Section 4 to give more information about the numerical cost of LIMA v2.0. It is not only the added complexity in the microphysics scheme that is responsible for this increase in the computation time. Additional processes in the microphysics scheme make the cpu time increase by 9.7% when LIMA v2.0 is used instead of LIMA v1.0. When the full 2-moment scheme is used, the additional numerical cost is mainly attributed to the increase in the number of prognostic variables that must be forced, transported (advection and turbulence), exchanged and stored...

13. *Section 3.1.1, In addition to (I)-(III), (IV) self-aggregation of snow, graupel are necessary. In this case, mixing ratio does not change but number concentration reduces. This point is important particularly for the prediction of snow.*

In LIMA v2.0, we have chosen to deal only with snow self-collection. Snow self-collection is treated as a special case of collisions between particles with non-negligible falling velocities (paragraph 4 in section 3.1.3).

14. Eq. (13)-(15) in Section 3.1.2, why is Exy excluded from the integration? Does the term depend on size or mass as was proposed by Böhm (1999)? Please clarify the formulation of Exy here.

Böhm, J. P., 1999: Revision and clarification of "A general hydrodynamic theory for mixed-phase microphysics." Atmospheric Research, 52, 167–176, https://doi.org/10.1016/S0169-8095(99)00033-2.

The expressions for Exy for the various processes are given in Section 3.1.2 (lines 229 to 245 in the first version of the manuscript). They do not depend on the characteristics of the hydrometeors, but are simply constant or function of temperature.

15. Eq. (13)-(15) in Section 3.1.2, what is $\rho$dref ? Based on Eq. (3), that is to be $\rho$a .

You are right: $\rho$dref has been changed to $\rho$a in the revised manuscript.

16. Eq. (16) contains undefined terms of $\Delta$DRYG r c , $\Delta$DRYG r r , $\Delta$ DRYG r i , and $\Delta$DRYG r g. I suppose the terms as $\Delta$COL r c (c − g), $\Delta$COL r r (r − g), $\Delta$COL r i (r − i), and $\Delta$COL r s (g − s) as was defined in the case (II) in Section 3.1.1. Is it right?

A list of symbols has been added as an appendix in the revised version of the manuscript. The terms are also defined in the text in the revised version of the manuscript.

17. Title of Section 3.1.3., the word "significant" would be replaced with "non-negligible". Similarly, significant at line#247 would be done.

Done.

18. Line#247, please close the sentence before "and" to increase readability. Then, please start the sentence with "Therefore" instead of using "therefore" in the middle part of the sentence. In addition, it would be better to insert "in this study" at the end of the sentence.

Done.

19. Line#265-273 (Rain accretion on aggregates ACC), do the diameters mean the diameter of individual particles or PSD mean diameter? If it means the diameter of individual particles, how do you integrate the collection kernel? Could you clarify this point?

Drlim is the mean diameter above which raindrop-collecting aggregates are considered as graupels. This is clarified in the text.

20. Line#285, what is "a threshold"? Please clarify that.

The conversion of hail into graupel occurs when cloud droplets and hail mixing ratios decrease below 0.001 g kg-1 and 0.01 g kg-1, respectively. This is clarified in the new version of the scheme.

21. Line#292, "primary ice crystal" would be a typo of "pristine ice crystal".

This has been corrected in the new version of the manuscript.

22. Line#292, $\Delta CNV\,r\,i$ has not been documented. Please document the equation of the growth term or please refer to the original article in which the term is documented.

This equation does not change between the v1.0 and v2.0 schemes. Therefore, a reference to Vié et al. (2016) has been added in the text for this process rate.

23. Line#299, water formed "on" the surface

Done.

24. Line#302, here the authors assumed melting graupel particles are larger than 0.72 mm. Do you use something of a criterion for shedding graupel diameter?

There is no criterion for shedding graupel diameter.

25. Line#303-304, please refer to the articles, which document the wind-tunnel experiments.

The reference to Mitra et al. (1990) has been added in the revised version of the manuscript.

26. Line#361, At first glance, I don't understand what do "30 (3.5 mm/h)" means. To increase readability, it is better to modify as "30 min (3.5 mm/h), 34 min (4.5mm/hr) ...".

Done.

27. Line#398, this sentence is not necessary.

This sentence has been removed.

28. Line#412, you should remove "it seems to be" because this point is evident from the figure. In addition, it is better to mention the difference in snow amount. Since snow is produced by aggregation of cloud ice, large amount of snow indicates the rapid consumption of cloud ice. This point is clearly shown by Figure 8e.

"It seems to be" has been replaced by "it is".

Yes, in the manuscript, we mentioned that: "Figure 8e shows that the aggregation of ice crystals on snow is less effective when the concentration is prognostic (LIMA2)."

29. Line#420, after "30 minutes" would be "40 minutes".

You are right: this has been modified in the revised version of the manuscript.

30. Line#423, what hypothesis do you mention here. I guess that is an issue in the number diagnosis. Please clarify that.

Yes, it refers to the hypothesis in the number concentration diagnosis in the 1-moment scheme. This is clarified in the revised version of the manuscript.

31. *Line#452, In general, "not shown" is used when it is not important and it does not change conclusion. When the authors did not show the figure, the results were not verified. Thus, please do not use "verified" here. I suggest to remove the sentence or add the figure. I think that point was found in Figures 8a-d, so you can refer to the figure in this sentence.*

This sentence has been removed. Figure 8 shows averaged vertical profiles for the transfer rates between 14 and 24 minutes, so it does not show the evolution of the processes all along the simulation.

32. *Line#462, I don't understand the wording "for the benefit" here. Isn't it deficient? Wrong diagnosis of the number concentrations results in wrong estimation of radiative properties. I think this is the deficient in 1-moment schemes.*

This sentence has been modified.

33. *Line#502, the wording "decoupling" does not match the context because the number concentration and mixing ratio should be coupled through the physical processes as was represented by LIMA2. Please change the wording.*

This sentence has been modified.

**References**

Caniaux, G.: Paramétrisation de la phase glace dans un modèle non hydrostatique de nuage : application à une ligne de grains tropicale. 257 pp., PhD thesis, Paul Sabatier University, Toulouse 3, 1993.

Mitra, S. K., O. Vohl, M. Ahr, and H. R. Pruppacher, 1990: A Wind Tunnel and Theoretical Study of the Melting Behavior of Atmospheric Ice Particles. IV: Experiment and Theory for Snow Flakes. *J. Atmos. Sci.*, **47**, 584–591, https://doi.org/10.1175/1520-0469(1990)047<0584:AWTATS>2.0.CO;2.

Pinty, J. and Jabouille, P.: A mixed-phase cloud parameterization for use in mesoscale non hydrostatic model: simulations of a squall line and of orographic precipitations. In: Proceedings of Conference on Cloud Physics, Everett, WA, pp. 217–220. American Meteorological Society, Boston, MA, 2018.

Vié, B., Pinty, J.-P., Berthet, S., and Leriche, M.: LIMA (v1.0): A quasi two-moment microphysical scheme driven by a multimodal population of cloud condensation and ice freezing nuclei, Geosci. Model Dev., 9, 567–586, https://doi.org/10.5194/gmd-9-567-2016, 2016.

---

## Referee Report (RR1)

The reviewers have satisfactorily addressed my comments, and I have only a few minor technical corrections to suggest prior to publication. I reference line numbers in the "track changes" version of the manuscript.

**Minor comments**:

- It is good that you reference P3 as another type of scheme in L54-59, but I believe you mistakenly lump it together with MG08 which is in fact category and moment based. I would remove the citation for MG08. Also please clarify in the text that by "LIMA-like" (L57), I assume you mean category-based?
- Please add the units (where appropriate) for the parameters listed in Table 1. The list of symbols that has been added to the appendix is very helpful.
- Figures 2. 4, 5 and others are much improved over the original version.
    - The caption of figure 4 should read "Instantaneous rain precipitation (**shading**)" so as not to confuse the readers with the shading versus colored contours.
    - The caption of figure 5 has the wrong colors listed for snow, graupel, and hail compared with the legend in the figure.
    - The yellow lines in Figure 8 are challenging to see on a screen and do not show up when printed in black and white. I suggest choosing a darker color like magenta.
- In order to address the 2nd reviewer's comment on distinguishing between LIMA and LIMA2, I would further suggest writing out "LIMA1" everywhere that the v1 model is the subject of discussion, rather than the more generic "LIMA".

**Technical corrections:**

L47: "the concern to develop" → "concern about developing"

L75: "to include" → "of including"

L133: "Gamma distribution law" → "Gamma distribution" (law is redundant)

L151: "broadening of particle" → "broadening of the particle"

L151: "verified" → "met"

L485: "diagnostics of Ns is not…" → "diagnostics of Ns are not…"

---

## Referee Report (RR2)

Review of "LIMA (v2.0): A full two-moment cloud microphysical scheme for the mesoscale non-hydrostatic model Meso-NH v5-6" by Tanfour et al."

The revised manuscript thoroughly described the modeling method of the 2-moment bulk cloud microphysics scheme and the advantage of 2-moment scheme in comparison to the basic 1-moment scheme. The manuscript is well documented. Therefore, my recommendation is acceptable after revising a few technical issues.

Technical comments

1. Figure 5.

   I don't recognize graupel as green color. Isn't it yellow? In addition, hail seems to be purple whereas hail is indicated as black in the caption. Please correct them.

2. Figure 8.

   The figure caption is confusing. What do you mean "to produce Fig.6" ?

3. Line# 781, What are the "constants"? Please clarify them.

4. Line# 585, please never use "seems to be" in the conclusion section. That expression is too weak. It is better to remove the sentence or reword them to present clear relationships found from the results.

---

## Author Response (AR2)

**Technical corrections**

- Figure 8: The figure caption is confusing. What do you mean "to produce Fig.6" ? The figure caption has been modified.

- Line# 781, What are the "constants"? Please clarify them. Done

- Line# 585, please never use "seems to be" in the conclusion section. That expression is too weak. It is better to remove the sentence or reword them to present clear relationships found from the results. This sentence has been modified.

- In L54-59, P3 is likely mistakenly lumped together with MG08 which is in fact category and moment based. Suggest removing the citation for MG08. Also please clarify in the text that by "LIMA-like" (L57), I assume you mean category-based? The reference to MG08 has been removed. The expression LIMA-like has been removed and replaced by « based on predefined categories ».

- Please add the units (where appropriate) for the parameters listed in Table 1. Done.

- The caption of figure 4 should read "Instantaneous rain precipitation (shading)" so as not to confuse the readers with the shading versus colored contours. Done.

- The caption of figure 5 has the wrong colors listed for snow, graupel, and hail compared with the legend in the figure (is graupel shown with green color? Isn't it yellow? In addition, hail seems to be purple whereas hail is indicated as black in the caption) Done.

- The yellow lines in Figure 8 are challenging to see on a screen and do not show up when printed in black and white, suggest choosing a darker color like magenta. The yellow lines have been replaced by vermilion lines in Figure 8.

- suggest writing out "LIMA1" everywhere that the v1 model is the subject of discussion, rather than the more generic "LIMA". Done. Figures labels have also been modified accordingly.

- L47: "the concern to develop" -> "concern about developing". Done.

- L75: "to include" -> "of including". Done.

- L133: "Gamma distribution law" -> "Gamma distribution" (law is redundant). Done.

- L151: "broadening of particle" -> "broadening of the particle". Done.

- L151: "verified" -> "met". Done.

- L485: "diagnostics of Ns is not…" -> "diagnostics of Ns are not… Done.

- replace raster graphics in Fig. 1 with vector graphics. Done.

- fix DOI URL in Khain et al. 2000 (https://doi.org/https://…). Done.

- fix DOI URL in Brown et al. 2017 (https://doi.org/https://…). Done.

- add URL to Caniaux 1993: https://theses.fr/1993TOU30071. Done.

- use journal name abbreviations in all journal-article reference entries (acronym database suggested by GMD:

  https://images.webofknowledge.com/images/help/WOS/A_abrvjt.html). Done.